# VIBE CHECKER: ALIGNING CODE EVALUATION WITH HUMAN PREFERENCE

## ABSTRACT

Large Language Models (LLMs) have catalyzed vibe coding, where users leverage LLMs to generate and iteratively refine code through natural language interactions until it passes their *vibe check*. *Vibe check* is tied to real-world human preference and goes beyond functionality: the solution should feel right, read cleanly, preserve intent, and remain correct. However, current code evaluation remains anchored to pass@k and captures only functional correctness, overlooking the non-functional instructions that users routinely apply. In this paper, we hypothesize that instruction following is the missing piece underlying *vibe check* that represents human preference in coding besides functional correctness. To quantify models' code instruction following capabilities with measurable signals, we present VERICODE, a taxonomy of 30 verifiable code instructions together with corresponding deterministic verifiers. We use the taxonomy to augment established evaluation suites, resulting in VIBE CHECKER, a testbed to assess both code instruction following and functional correctness. Upon evaluating 31 leading LLMs, we show that even the strongest models struggle to comply with multiple instructions and exhibit clear functional regression. Most importantly, *a composite score of functional correctness and instruction following correlates the best with human preference*, with the latter emerging as the primary differentiator among advanced LLMs on real-world programming tasks. Our work identifies core factors of the *vibe check*, providing a concrete path for benchmarking and developing models that better align with user preferences in coding.

## 1 INTRODUCTION

Large Language Models (LLMs) have reshaped how humans write code, fostering a workflow termed "*vibe coding*" (Karpathy, 2025; Willison, 2025). In this paradigm, AI's role shifts from a one-shot code completion tool for developers to an interactive collaborator for a broader audience, including users with limited coding experience. Through multi-turn natural language interactions, users can create and refine solutions from scratch, requiring the model to maintain context, adapt to evolving requirements, and iteratively improve the code until it meets their needs (Ross et al., 2023; Yang et al., 2023). The user's final accept/reject choice serves as a real-time evaluation: what we call the "*vibe check*," a subjective preference typically based on whether the solution feels right, reads cleanly, avoids obvious issues or anti-patterns, and preserves intent and correct functionality. This collaborative workflow, popularized by tools such as Copilot[1] and Cursor[2], is rapidly becoming standard practice in modern software development (Peng et al., 2023; Stack Overflow, 2025).

Despite the shift toward vibe coding, existing code evaluation remains anchored to functional correctness, typically measured as pass@k (Chen et al., 2021; Austin et al., 2021; Jimenez et al., 2024). These metrics indicate whether code passes unit tests but abstract away non-functional expectations that users apply when selecting a response, including adherence to project conventions, documentation clarity, minimal and targeted edits, and preservation of prior intent across interactions. This disconnection is evident in platforms such as Copilot Arena (Chi et al., 2025), a large-scale vibe-checking scenario where human programmers choose preferred candidate snippets. Strikingly, rankings of code LLMs from Copilot Arena exhibit weak or negative correlations with functional scores

---

[1] https://github.com/features/copilot
[2] https://cursor.com

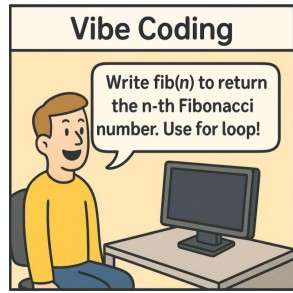 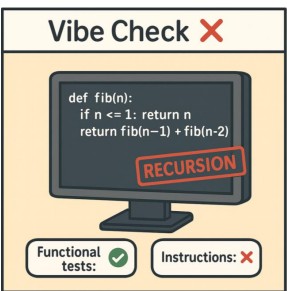 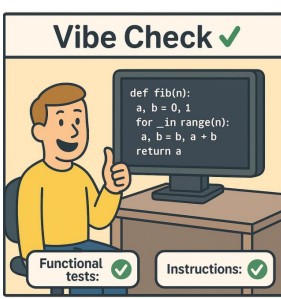

Figure 1: *Vibe check* goes beyond functionality, requiring code to satisfy non-functional instructions such as coding style and logic patterns, which are also key factors of human preference.

on popular benchmarks. Moreover, pass@k remains a dominant verifiable reward signal in RLVR training (DeepSeek-AI, 2025; Da et al., 2025), steering optimization toward an incomplete notion of code quality. Consequently, models can achieve high leaderboard scores yet fail the vibe check in practice, producing code that is technically correct but misaligned with user preferences.

To bridge this gap, we hypothesize that the non-functional signals emerging from interactions are an important, yet under-measured, component of the vibe check. We first introduce VERICODE, a taxonomy of verifiable code instructions designed to capture what users routinely screen for during code selection. Grounded in hundreds of rules from industrial linters and style guides, we perform manual curation and automated filtering to distill a core set of 30 instructions across five categories. Each instruction is paired with a verifier implemented using standard linters and abstract syntax tree analysis. These verifiers yield a binary pass or fail score, enabling reliable automatic evaluation while also providing a verifiable and scalable reward source for model training.

Building on VERICODE, we augment established benchmarks, BigCodeBench (Zhuo et al., 2025) and LiveCodeBench (Jain et al., 2025), with these verifiable instructions to better simulate real-world interactions. We refer to the augmented variants as BigVibeBench and LiveVibeBench. For each user query, an LLM-driven selector chooses a relevant and non-conflicting subset of instructions from our taxonomy to add as explicit constraints. Functional unit tests together with our instruction verifiers constitute a unified testbed, VIBE CHECKER, which measures both functional correctness and instruction following (IF). Using this testbed, we evaluate 31 LLMs from 10 model families in two realistic settings: single-turn generation, in which the model must satisfy all constraints in one pass, and multi-turn editing, in which constraints are introduced sequentially while preserving prior intent. This setup allows us to study both dimensions across interaction contexts.

Our analysis on VIBE CHECKER testbed yields several key insights into the code evaluation:

- **Non-functional instructions cause notable functional regression.** Although the added instructions do not target functionality, pass@1 decreases across all models. Under five instructions, average pass@1 drops by 5.85% and 6.61% on the two augmented benchmarks (Section §4.2).

- **Following multiple instructions remains challenging for LLMs.** Even the best performing model reaches only 46.75% and 40.95% success rate under five instructions on BigVibeBench and LiveVibeBench (Section §4.3). Models also exhibit a position bias for instruction following, with mid-position instructions followed less reliably than those at the beginning or end (Section §4.4).

- **Single-turn vs. multi-turn interactions alter LLM behavior.** Under the same tasks, single-turn generation better preserves functionality but follows fewer instructions, whereas multi-turn editing achieves higher IF at the cost of more functional regressions (Sections §4.2 and §4.3).

- **Human preference reflects a mixture of functional correctness and instruction following.** On the coding subset of LMArena (Chiang et al., 2024), a composite of functional correctness and our IF score correlates better with model ratings than either metric alone, with IF emerging as the key differentiator among advanced models on the real-world programming tasks (Section §4.5).

In summary, this work establishes IF as an essential, yet overlooked, component of code evaluation. Our VERICODE taxonomy and VIBE CHECKER testbed offer a concrete path to benchmark and develop models against a more human-aligned notion of code quality beyond functionality.

## 2 VERICODE: A TAXONOMY OF VERIFIABLE CODE INSTRUCTIONS

To quantify the IF capability, we first construct VERICODE, a taxonomy of verifiable code instructions. This section presents its design principles, construction process, and resulting structure.

### 2.1 DESIGN PRINCIPLES

We design VERICODE around four core principles to ensure it is rigorous, relevant, and useful:

- **Verifiability.** Each instruction is paired with an automated, deterministic verifier that returns a binary pass/fail signal, enabling objective and scalable evaluation.
- **Practice Grounding.** Instructions reflect common developer expectations and conventions, drawing on widely used standards rather than synthetic or adversarial constraints.
- **Comprehensive Coverage.** The set spans key non-functional aspects, including coding style, logic patterns, documentation, error handling, and API or library constraints.
- **Difficulty.** Instructions are curated to be meaningfully challenging and diagnostic, ensuring that recent advanced LLMs exhibit imperfect adherence.

### 2.2 TAXONOMY CONSTRUCTION PROCESS

We carefully curate VERICODE in three stages: sourcing a candidate pool, performing multi-stage filtering, and finalizing the set with expert review and verifier implementation.

**Candidate Pool Sourcing.** We source our initial candidate pool from Ruff, an industry-standard Python linter that aggregates more than 800 rules drawn from popular tools[3]. This provides a high-coverage inventory of practices that users routinely follow and check. Static linting, however, inspects only code and cannot evaluate instructions that target the entire response (e.g., append a JSON explanation after the code block). To close this gap, we add a set of instructions focusing on documentation outside the code blocks, extending coverage beyond what static analysis can capture.

**Scope and Relevance Filtering.** The initial pool is first filtered for scope and relevance. We apply a top-down consolidation to address rule overlap, prioritizing broader instructions over their more specific subsets. This stage ensures that each instruction is broadly applicable across common coding tasks and not confined to niche scenarios.

**Difficulty Filtering.** We then screen the remaining candidates for difficulty. Using Gemini 2.5 Flash (Gemini Team, 2025) on a challenging test set, BigCodeBench-Hard (Zhuo et al., 2025), we measure instruction following rate alongside functional correctness at pass@1. Any instruction with a success rate above 90% and no degradation in pass@1 is removed. Borderline cases are flagged for manual review. This step focuses on non-trivial constraints that challenge advanced LLMs.

**Review and Verifier Implementation.** The final instruction set is manually reviewed by domain experts on the author team with coding-research experience to ensure clarity and real-world relevance. For verification, we prioritize linter-backed checks when available and implement deterministic tests using Abstract Syntax Tree (AST) analysis and regular expressions when no direct rule exists. All verifiers share a common interface: a testing function that returns a binary pass or fail, enabling scalable evaluation and reproducibility.

### 2.3 RESULTING VERICODE TAXONOMY

The multi-stage construction process yields our final verifiable taxonomy VERICODE[4].

**Taxonomy Structure.** The final set contains 30 instructions organized into five categories: *Coding Style & Conventions (9)*, *Logic & Code Patterns (9)*, *Documentation & Commenting (6)*, *Error Handling & Exception Management (4)*, and *Library & API Constraints (2)*. The taxonomy is organized hierarchically: the root represents the overall concept of verifiable code instructions, the five categories form the top-level nodes, and the 30 individual instructions are the leaf nodes. Our current

---

[3]https://docs.astral.sh/ruff/rules
[4]We will publicly release the taxonomy together with the corresponding verifiers to support community use.

| Category | Prompt | Verifier | Parameter |
|----------|--------|----------|-----------|
| **Coding Style & Conventions** | Write code ensuring all lines are no longer than {`line_length`} characters. | E501 Rule | `line_length` (int) Recommended: 79 (classic), 88 (modern) |
| **Logic & Code Patterns** | Ensure each function has at most {`max_branches`} branches. | PLR0912 Rule | `max_branches` (int) Recommended: 2–4 |
| **Documentation & Commenting** | Document your code using the {`convention`} docstring format. | D Rule | `convention` (str) Supported: Google, NumPy, PEP 257 |
| **Error Handling & Management** | Replace all aliases with the canonical *OS-Error* exception. | UP024 Rule | None |
| **Library & API Constraints** | Replace all *os*, *os.path*, *glob*, and *open* with their *pathlib* equivalents. | PTH Rule | None |

Table 1: Refined examples from VERICODE taxonomy. Each instruction maps to a verifiable linter rule and includes tunable parameters where applicable. Full versions are provided in Appendix B.2.

instantiation focuses on Python, the dominant language in code evaluation, but the framework is language-agnostic and can be applied to other languages using standard linters.

**Instruction Schema.** Each instruction specifies five necessary elements: 1) category, 2) description, 3) distinct prompts for both single-turn generation and multi-turn editing, 4) configurable *parameters* with recommended or supported values, and 5) the verification code that returns a binary score. A full version of the instructions is available in Appendix B.2.

A key feature of our taxonomy is its extensibility, which is achieved through the *Parameters* field. As illustrated in Table 1, parameters such as `line_length`, `max_branches`, or documentation conventions allow a single instruction to generate multiple variants with different difficulty levels. This flexibility enables our set of 30 core instructions to be programmatically expanded into hundreds of distinct and checkable constraints, providing a scalable framework for future research.

## 3  VIBE CHECKER: A NEW TESTBED FOR CODE EVALUATION

Building on proposed VERICODE, we introduce VIBE CHECKER – a testbed that extends standard code benchmarks with explicit, verifiable instructions. It evaluates models under both single- and multi-turn protocols, measuring functional correctness as well as instruction following capabilities.

### 3.1  BENCHMARK AUGMENTATION

We ground our evaluation in established benchmarks, which allows us to leverage their unit tests to consistently measure functional correctness and situate our analysis within widely used evaluation suites. Concretely, we construct two augmented variants:

- **BigVibeBench**, adapted from BigCodeBench to cover real-world programming tasks.
- **LiveVibeBench**, adapted from LiveCodeBench to cover algorithmic/contest problems.

This combination ensures that our evaluation covers a diverse range of coding challenges. Our augmentation process involves the following stages:

**Instruction Selection.** For each user query, we randomly permute the full set of 30 taxonomy instructions to form an ordered list. An LLM-based selector then scans this permuted list once, deciding whether to keep or discard each instruction based on two criteria: 1) *Relevance*: the instruction must pertain to the query and plausibly influence the implementation, and 2) *Non-conflict*: the instruction must not contradict any instruction already selected earlier in the pass. The accepted instructions, in this permuted order, constitute the constraint set used to evaluate all models.

**Parameter Selection and Validation.** Once the instructions are selected, we prompt an LLM to assign specific parameter values to each one. To guide this generation, the prompt includes the supported keys, types, ranges, and recommended values in our taxonomy, as well as the context of the user query, aiming for parameters that are both achievable and challenging. Finally, the generated

Figure 2: Our evaluation protocol simulates two real-world interaction patterns: *single-turn generation*, where all instructions are given upfront in one prompt, and *multi-turn editing*, where instructions are introduced sequentially to refine a solution. Both are measured for functionality and IF.

parameters undergo a rule-based validation step: any parameter keys not explicitly defined for that instruction are removed, and any invalid values are reverted to predefined defaults.

Both Gemini 2.5 Pro (Gemini Team, 2025) and Claude 4 Opus (Anthropic, 2025) are tested as selectors in our augmentation pipeline, yielding similar instruction-category distributions. The final benchmark is augmented by Claude 4 Opus, chosen for its lower invalid-parameter rate (0.96% vs. 2.47% for Gemini 2.5 Pro). The resulting distributions show that instructions for *Coding Logic*, *Coding Style*, and *Documentation* are most prevalent, with *Coding Logic* being particularly frequent in the algorithm-focused LiveCodeBench (see Figure 12 in the Appendix for a full breakdown).

## 3.2 EVALUATION PROTOCOL

Our evaluation protocol, illustrated in Figure 2, mirrors real-world usage by providing single- and multi-turn interactive settings with two evaluation metrics.

**Interactive Settings.** We use two settings that differ in how instructions are presented:

- *Single-Turn Generation* presents all selected instructions after the original query within one prompt. The model returns a single implementation.
- *Multi-Turn Editing* first elicits an initial implementation in response to the original query, then reveals the selected instructions one at a time. At each round, the model sees the full interaction history and updates the solution. The code from the last round is used for evaluation.

**Evaluation Metrics.** For both settings, we evaluate the code on two axes:

- *Functionality:* We measure functional correctness with unit tests and report functional regression $\mathrm{FR}_k$ from adding $k$ instructions. Let $S_k$ denote the functional score (typically pass@1) after injecting $k$ instructions, with $S_0$ the score on the original prompt. The rate is calculated as:

$$\mathrm{FR}_k = \frac{S_0 - S_k}{S_0}.$$

- *Instruction Following:* We report IF at two granularities. For a task with $k$ instructions, let $I_j \in \{0, 1\}$ indicate whether instruction $j$ passes its verifier. The *instruction-level* score averages per-instruction passes, and the *task-level* score requires all passes:

$$\mathrm{IF}_{\mathrm{instruction}} = \frac{1}{k} \sum_{j=1}^{k} I_j, \qquad \mathrm{IF}_{\mathrm{task}} = \mathbb{1}\left[\sum_{j=1}^{k} I_j = k\right].$$

Here, a task refers to a benchmark problem together with its selected instruction set.

| Model | | Single-Turn Generation ↓ | | | | | Multi-Turn Editing ↓ | | | | |
|---|---|---|---|---|---|---|---|---|---|---|---|
| | Base | 1 Inst | 2 Inst | 3 Inst | 4 Inst | 5 Inst | 1 Inst | 2 Inst | 3 Inst | 4 Inst | 5 Inst |
| **BigVibeBench: Real-World Programming Tasks** | | | | | | | | | | | |
| Gemini 2.5 Pro | 50.35 | 0.34 | 2.60 | 0.87 | -0.36 | 1.39 | 1.75 | 2.44 | 4.01 | 4.89 | 5.04 |
| Gemini 2.5 Flash | 47.37 | 0.74 | 1.12 | 2.60 | 1.31 | 2.41 | 0.93 | 1.12 | 1.48 | 2.98 | 3.72 |
| Claude 4 Opus | 51.05 | -0.86 | -2.23 | -4.31 | -1.72 | -2.08 | 0.51 | 1.02 | 2.06 | 3.25 | 3.78 |
| Claude 4 Sonnet | 51.84 | -0.17 | -0.52 | 0.33 | 0.50 | 0.50 | 0.85 | 2.03 | 3.55 | 4.05 | 5.40 |
| GPT 5 | 46.49 | 0.56 | 5.66 | 2.26 | 3.20 | 1.89 | 1.70 | 2.82 | 4.35 | 5.27 | 5.46 |
| o4 mini | 52.28 | 4.02 | 9.39 | 5.87 | 7.38 | 9.56 | 2.18 | 4.71 | 7.04 | 7.04 | 8.05 |
| Kimi K2 | 47.19 | -1.12 | -0.19 | -0.93 | 0.17 | 2.03 | 2.23 | 4.09 | 2.78 | 4.45 | 6.12 |
| **LiveVibeBench: Algorithmic Programming Contest Problems** | | | | | | | | | | | |
| Gemini 2.5 Pro | 85.31 | -0.11 | 3.45 | 2.45 | 2.45 | 2.45 | 0.67 | 1.34 | 1.01 | 1.89 | 2.23 |
| Gemini 2.5 Flash | 74.50 | 3.56 | 5.34 | 8.01 | 5.60 | 6.74 | 0.12 | 1.14 | 1.65 | 3.44 | 3.69 |
| Claude 4 Opus | 68.72 | 4.55 | 8.56 | 8.41 | 8.13 | 8.96 | 2.07 | 1.38 | 1.51 | 2.34 | 2.34 |
| Claude 4 Sonnet | 66.35 | 4.57 | 5.00 | 3.71 | 6.99 | 9.00 | 0.42 | 0.86 | 1.15 | 1.72 | 2.14 |
| GPT 5 | 71.47 | 1.72 | 2.13 | 3.32 | 7.16 | 6.76 | 2.25 | 4.24 | 5.57 | 7.43 | 9.02 |
| o4 mini | 80.95 | 5.74 | 9.02 | 9.02 | **11.37** | **12.29** | 3.63 | 8.91 | **10.19** | **11.71** | **15.92** |
| Kimi K2 | 63.58 | 8.92 | **15.48** | **16.07** | **15.48** | **16.36** | 2.64 | 5.63 | 9.50 | **12.49** | **12.79** |

Table 2: Top-performing models still suffer from functional regression when non-functional instructions are added. *Base* is pass@1 on the original query. All other columns report the regression rate (%) relative to *Base*. $k$ Inst is the number of added instructions. Light red marks $> 5\%$ regression and **deep red** denotes $> 10\%$. Full results for all 31 LLMs are listed in the Appendix D.2.

# 4 EXPERIMENTS

Based on VIBE CHECKER, this section investigates the trade-off between functionality and instruction following, analyzes LLM behaviors, and ultimately correlates our metrics with user preference.

## 4.1 EXPERIMENTAL SETUP

**Models.** To ensure a comprehensive analysis, we select a cohort of 31 powerful LLMs spanning 10 distinct model families, including Gemini (Gemini Team, 2025), Claude (Anthropic, 2024; 2025), OpenAI (OpenAI, 2024; 2025), DeepSeek (DeepSeek-AI, 2024; 2025), Qwen (Hui et al., 2024; Qwen Team, 2025), Grok (xAI, 2025a;b), Gemma (Gemma Team, 2025), Mistral (Mistral AI, 2025), MiniMax (MiniMax, 2025), and Kimi (Kimi Team, 2025).

**Benchmarks.** We construct BigVibeBench and LiveVibeBench by augmenting the full sets of Big-CodeBench (1,140 instances) and LiveCodeBench v1–v6 (1,055 problems, May 2023 to May 2025). Each instance across both benchmarks is augmented with 5 instructions from VERICODE taxonomy, resulting in a total of over 10K instruction-level evaluations.

**Implementation Details.** All models are queried via the Vertex AI[5] and OpenRouter[6] APIs. During benchmark augmentation, we use a deterministic temperature of 0.0. During evaluation, we follow the defaults of the underlying benchmarks: 0.0 for BigVibeBench and 0.2 for LiveVibeBench. We enable thinking mode on all models that support it. For Claude models with thinking mode enabled, the API requires temperature 1.0, so we set it accordingly; all other models use the benchmark defaults. The context length is set to each model's supported maximum, capped at 32,768 tokens.

## 4.2 RESULTS FOR FUNCTIONALITY

**Adding non-functional instructions leads to functional regression.** Table 2 reports regression rates on BigVibeBench for real-world programming and LiveVibeBench for algorithmic problems. Handling multiple non-functional instructions is routine in practice, yet it still causes notable functional loss even for state-of-the-art models. On BigVibeBench, under multi-turn editing with five

---

[5]https://cloud.google.com/vertex-ai/docs/reference/rest
[6]https://openrouter.ai

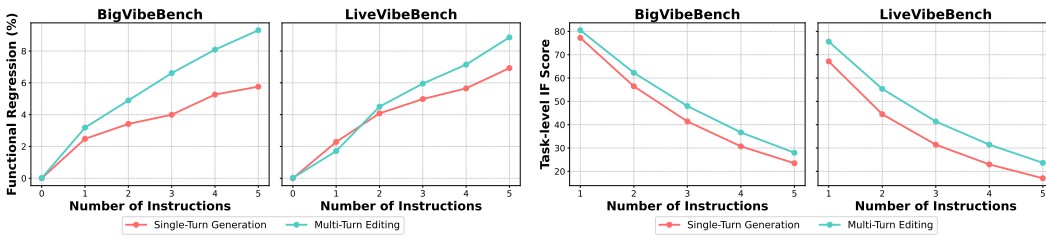

(a) Functional Regression Rate.

(b) Task-Level Instruction Following.

Figure 3: Trends averaged over all evaluated models. As the number of instructions increases, functional regression grows steadily, while the task-level IF score drops markedly. Single-turn generation better preserves functionality, whereas multi-turn editing achieves higher instruction following.

| Model | Single-Turn Generation ↑ | | | | | Multi-Turn Editing ↑ | | | | |
|---|---|---|---|---|---|---|---|---|---|---|
| | 1 Inst | 2 Inst | 3 Inst | 4 Inst | 5 Inst | 1 Inst | 2 Inst | 3 Inst | 4 Inst | 5 Inst |
| **BigVibeBench: Real-World Programming Tasks** | | | | | | | | | | |
| Gemini 2.5 Pro | 82.19 | 60.70 | 48.16 | 37.46 | 30.70 | 84.56 | 68.33 | 55.61 | 44.21 | 33.68 |
| Gemini 2.5 Flash | 81.67 | 61.05 | 43.68 | 30.53 | **25.70** | 78.68 | 56.75 | 40.96 | **29.12** | **21.75** |
| Claude 4 Opus | 88.77 | 76.32 | 64.21 | 52.98 | 46.75 | 87.02 | 73.16 | 61.05 | 51.32 | 42.11 |
| Claude 4 Sonnet | 84.91 | 67.19 | 52.28 | 42.98 | 35.26 | 86.40 | 72.54 | 61.23 | 51.05 | 42.89 |
| GPT 5 | 82.89 | 67.63 | 54.04 | 42.98 | 34.39 | 84.91 | 72.37 | 62.98 | 55.26 | 48.51 |
| o4 mini | 84.82 | 70.79 | 57.11 | 47.98 | 41.32 | 88.51 | 74.74 | 61.23 | 50.09 | 41.84 |
| Kimi K2 | 85.00 | 68.86 | 53.68 | 41.23 | 30.18 | 89.12 | 77.11 | 66.40 | 53.95 | 44.04 |
| **LiveVibeBench: Algorithmic Programming Contest Problems** | | | | | | | | | | |
| Gemini 2.5 Pro | 75.83 | 56.78 | 45.50 | 37.63 | **29.57** | 78.96 | 61.61 | 51.18 | 41.04 | 32.80 |
| Gemini 2.5 Flash | 66.54 | 45.97 | 32.89 | **23.03** | **17.06** | 72.80 | 51.09 | 34.98 | **25.31** | **17.82** |
| Claude 4 Opus | 78.86 | 57.91 | 47.96 | 38.96 | 35.17 | 85.59 | 72.89 | 61.71 | 52.04 | 43.70 |
| Claude 4 Sonnet | 75.73 | 56.40 | 44.17 | 35.36 | **28.53** | 84.45 | 73.46 | 62.37 | 52.70 | 44.64 |
| GPT 5 | 82.18 | 68.53 | 55.17 | 47.01 | 40.95 | 85.59 | 74.50 | 66.64 | 57.35 | 50.14 |
| o4 mini | 73.18 | 53.93 | 43.22 | 33.36 | **27.20** | 81.52 | 66.64 | 54.60 | 42.84 | 32.61 |
| Kimi K2 | 62.75 | 41.61 | **27.77** | **19.05** | **11.94** | 76.97 | 57.35 | 44.17 | 35.73 | **27.87** |

Table 3: Following multiple instructions remains challenging for top-performing models. We report the task-level IF scores on both benchmarks. Light red marks IF score < 50 and **deep red** indicates IF < 30. Full results for all 31 LLMs are provided in the Appendix D.3.

instructions, every model shows a regression above 5% except Gemini 2.5 Flash and Claude 4 Opus. The effect is amplified on LiveVibeBench: regressions above 5% occur frequently for all models except Gemini 2.5 Pro, with the impact particularly pronounced for o4 mini and Kimi K2, which exceed 10% in more than half of the test configurations.

**Single-turn generation better preserves functionality than multi-turn editing.** As illustrated in Figure 3a, regression increases monotonically with the number of instructions. On BigVibeBench, average regression for single-turn climbs from 2.48% with one instruction to 5.76% with five, while multi-turn rises from 3.18% to 9.31% over the same range. On LiveVibeBench, the gap is smaller: with two instructions, the two interaction modes are comparable, but as constraints increase, the single-turn setting gradually opens a clearer lead. Overall, single-turn generation more reliably preserves functionality, and its advantage grows with the number of instructions.

### 4.3 RESULTS FOR INSTRUCTION FOLLOWING

**Task-level success collapses under multiple instructions.** Table 3 presents the task-level IF score, where success requires satisfying all constraints simultaneously. The performance decay is rapid: with three or more instructions, most advanced models fall below 50 across both benchmarks. The decline is sharper on LiveVibeBench, where 5 of the 7 leading models do not reach 30 in the single-turn setting. Such a steep drop is not entirely unexpected, as even the best models remain below

90 on a single instruction. With each added instruction, the probability of satisfying all constraints decreases multiplicatively, yielding an exponential decay in task-level success. Such performance degradation indicates that IF remains a challenge for state-of-the-art models and should be prioritized in both evaluation and training to meet the demands of real-world, multi-instruction scenarios.

**Multi-turn editing is more effective for following instructions.** In contrast to the functionality results, multi-turn editing consistently outperforms single-turn generation in instruction following, as shown in Figure 3b. On BigVibeBench, the multi-turn setting maintains a 3% to 4.5% advantage in the task-level IF score. This gap widens on LiveVibeBench, where the advantage reaches around 8%. Given that the tasks are identical across settings, the consistent gap plausibly reflects the difference between the interactive patterns: single-turn must integrate all constraints in one pass and tends to prioritize preserving overall correctness, whereas the iterative nature of multi-turn supports targeted revisions that better satisfy newly introduced instructions.

### 4.4 INSTRUCTION POSITION ANALYSIS

**Models exhibit position bias in instruction following.** We define *instruction position* as the index of each constraint: for single-turn generation, the number in the list appended to the base prompt; for multi-turn editing, the round in which the constraint is introduced, starting at 1. On BigVibeBench, Figure 4 shows a clear U-shape, the classic "lost-in-the-middle" pattern typically reported for long-context generation (Liu et al., 2024), despite our prompts being only a few hundred tokens long. Furthermore, single-turn generation shows a primacy bias, performing best on the first instruction, while multi-turn editing displays a clear recency bias, peaking on

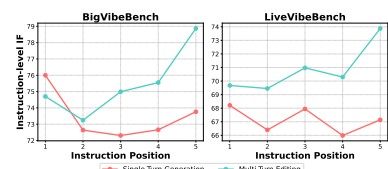

Figure 4: Average instruction-level IF trends by instruction position.

the final position. While the distinct U-shape does not generalize to the algorithmic tasks in LiveVibeBench, the underlying positional preferences remain consistent: single-turn generation favors the first instruction, while multi-turn editing consistently performs best on the last.

### 4.5 CORRELATING WITH HUMAN PREFERENCE

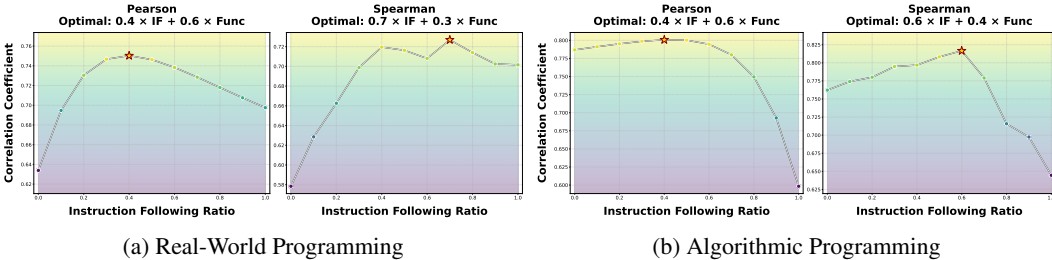

(a) Real-World Programming      (b) Algorithmic Programming

Figure 5: Human preference aligns best with a mix of IF and functionality. We correlate LMArena coding Elo with a composite score $\alpha$ IF $+ (1-\alpha)$ Func, where $\alpha \in [0, 1]$ is the weight on IF (x-axis). The peak correlation (starred) for both benchmarks is achieved with a mixture of the two metrics.

Having established metrics for both functionality and instruction following, we now investigate how these signals relate to overall human preference in coding tasks.

To explore this, we use LMArena (Chiang et al., 2024), currently the largest source of human preference data for LLMs. Its coding subset alone contains over 800K human votes, aggregated into Elo ratings for each model[7]. We take the latest default Elo ratings from this subset (see Appendix Table 4) and compute correlations against two metrics derived from VIBE CHECKER: **Func**, defined as pass@1 on the original problems, and **IF**, taken from the single-turn setting under one instruction. We then evaluate a composite score $\alpha$ IF $+ (1-\alpha)$ Func with $\alpha \in [0, 1]$, and report correlations.

**Human preference correlates best with a mixture of instruction following and functionality.** Across both benchmarks, the peak correlation occurs at intermediate $\alpha$ (starred in Figure 5), indi-

---

[7]https://lmarena.ai/leaderboard/text/coding

cating that neither IF nor Func alone explains preference as well as their combination. Concretely, on BigVibeBench, the optimum for Pearson correlation places a 40% weight on IF ($\alpha = 0.4$), while for Spearman correlation, the weight on IF rises to 70% ($\alpha = 0.7$). The optimal blend for Live-VibeBench is remarkably similar. In all cases, the mixture outperforms either isolated metric by a clear margin. Additional correlation types and results with LMArena style control (Li et al., 2024) disabled are reported in Appendix E.2, with conclusions remaining consistent.

**Which single factor users value depends on the coding scenario.** While a mix is always best, the importance of each metric considered alone differs by the type of programming task. For the real-world programming tasks in BigVibeBench, instruction following plays a more critical role. On the Spearman correlation, pure IF ($\alpha = 1$) correlates over 0.1 points higher with human preference than pure Func ($\alpha = 0$). For algorithmic programming tasks in LiveVibeBench, the opposite is true: pure Func holds a clear advantage over pure IF. This suggests that for practical, day-to-day coding, users place a high value on a model's ability to adhere to non-functional instructions, whereas, in competitive programming scenarios, functional correctness is the paramount factor.

**Overall implication.** Our results provide evidence that instruction following is a critical, under-measured component of human preference in coding tasks. Beyond functional correctness, adherence to non-functional constraints offers a strong signal for distinguishing real-world utility. Consequently, integrating instruction following alongside functionality in both evaluation and training provides a practical path toward models that align more closely with real-world user preferences.

## 5 RELATED WORK

**Instruction Following.** Research in general instruction following focuses on stress-testing models with synthetic constraints (e.g., forced word repetition) and evaluates with either deterministic checkers (Zhou et al., 2023; Wang et al., 2025; Pyatkin et al., 2025) or LLM-as-a-judge (Jiang et al., 2024; Qin et al., 2024). A prevailing trend leverages large-scale, verifiable instructions to boost capabilities via post-training, such as SFT and RL (Wang et al., 2025; Pyatkin et al., 2025). In contrast, instructions in the coding domain are tied to practical software development, concerning aspects such as logic patterns, coding style, and library usage. Prior work is sparse, and existing benchmarks for such code instructions lack verifiability. They typically compare to ground truth with DiffBLEU (Singhal et al., 2024) or use LLM and human judgment (Yan et al., 2025), which is unreliable and hard to scale. To bridge this gap, our work introduces a taxonomy of verifiable code instructions, each paired with a binary verifier, enabling scalable evaluation and training.

**Code Evaluation.** Functional correctness dominates code evaluation: the generated code is run against unit tests, from snippet-level functions (Chen et al., 2021; Austin et al., 2021; Hendrycks et al., 2021; Du et al., 2023; Lai et al., 2023; Liu et al., 2023; Jain et al., 2025; Zhuo et al., 2025; Zheng et al., 2025) to repository-level tasks (Jimenez et al., 2024; Chowdhury et al., 2024; Mündler et al., 2024; Yang et al., 2025; Zhao et al., 2025; Zan et al., 2025; Zhang et al., 2025). Research on non-functional requirements is a relatively small branch of research, covering aspects like adherence to task-oriented instructions (Yan et al., 2025), runtime efficiency, maintainability, and security (Singhal et al., 2024). We move beyond evaluating these aspects in isolation. On top of VIBE CHECKER testbed, we systematically analyze the trade-off between functional correctness and instruction following, and provide evidence that human preference reflects a composite of both dimensions. This work aims to align the code evaluation with the real-world user preferences.

## 6 CONCLUSION

In this paper, we challenged the prevailing focus on functional correctness in code evaluation. We study the *vibe check* as a subjective judgment tied to real-world human preference and approximate it with measurable signals. We present VERICODE, a verifiable taxonomy of non-functional code instructions, and VIBE CHECKER, a testbed that augments established evaluation suites. Across 31 leading LLMs, a composite of functional correctness and instruction following predicts human preference substantially better than either metric alone. Our work calls for moving beyond pass@k and for optimizing both functional and non-functional qualities in future research for coding.

## REPRODUCIBILITY STATEMENT

**VERICODE.** VERICODE is the taxonomy of verifiable non-functional code instructions. Section 2 details the design process and categories of VERICODE. Appendix B provides all the necessities to reproduce VERICODE: we present not only the verification code, but also five concrete case studies that contain all the elements in each category: *category, description, generation prompt, parameters, notes, and verification code*. Following Section 2 while referring to the concrete examples in Section 2 is sufficient to reproduce VERICODE.

**VIBE CHECKER.** VIBE CHECKER is the testbed, which we augment existing evaluation suites from BigCodeBench and LiveCodeBench by proposing new evaluation metrics and deploying VERICODE. We show the definition and calculation of our evaluation metrics in Section 3.2. For experiments conducted on VIBE CHECKER, Section 4.1 contains all the details, including the temperature we use, context window, etc. Meanwhile, Appendix C displays the system instructions and evaluation prompts we adopt for BigVibeBench and LiveVibeBench.

**Code and Data Release.** While the paper provides all necessary details for reproducing both the taxonomy (VERICODE) and benchmark results (VIBE CHECKER), we recognize that code and data release will further facilitate community use. We plan to publicly release the taxonomy along with the corresponding verifiers. We will also make every effort to release the generated code outputs from all 31 LLMs evaluated in the paper.

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

**Appendix Table of Contents**

## A    LLM USAGE STATEMENT

In the preparation of this manuscript, we use LLMs (e.g., Gemini) only to assist with language polishing. Its function is strictly limited to improving grammar, correcting spelling, and optimizing phrasing for clarity and readability. The LLMs do not contribute to any substantive part of the research, such as ideation, literature review, data analysis, or the generation of core arguments and conclusions. All technical content, claims, and conclusions come from the authors. The authors review and approve the final text and take full responsibility for its accuracy and integrity. LLMs are not authors or contributors.

## B    VERICODE TAXONOMY

### B.1    VERIFICATION CODE WITH RUFF

Given that 27 of the 30 verifiers in our VERICODE taxonomy are implemented via Python linter Ruff, we present the helper function in Figure 6.

```python
def _run_ruff_check(response: str, *ruff_args: str) -> bool:
    """
    A generic helper function to run Ruff with specific arguments.
    Uses stdin to pass code content to avoid file I/O overhead.
    """
    if not shutil.which("ruff"):
        raise RuntimeError(
            "Ruff is not installed or not in the system's PATH. "
            "Please install it with 'pip install ruff'."
        )
    try:
        command = [
            "ruff", "check", "-",  # "-" means read from stdin
            *ruff_args,
            "--no-fix",
            "--force-exclude"
        ]
        result = subprocess.run(
            command,
            input=response,
            capture_output=True,
            text=True,
            encoding='utf-8'
        )
        return result.returncode == 0
    except Exception:
        return False
```

Figure 6: Implementation of the core helper function used to run Ruff checks within VERICODE.

## B.2 CASE STUDIES FROM VERICODE

The full version of 5 instructions in Table 1 are presented in Figures 7, 8, 9, 10, and 11.

---

**ID: style_3**       **Category:** Coding Style & Conventions

---

**Description**

Enforce a maximum line length on the code, breaking long lines into multiple shorter lines to improve readability and conform to a specific constraint.

**Generation Prompt**

*Write code ensuring all lines are no longer than {line_length} characters.*

**Edit Prompt**

*Review the code and break any lines that are longer than {line_length} characters to ensure everything fits within that limit.*

**Parameters**

line_length: int = 79

**Notes**

This check verifies that all lines of code are at or below a given length using the `pycodestyle` rule `E501`. To test for compliance with common conventions, a recommended value is in the 79-88 character range (default: 79). This range covers the classic PEP 8 standard for code (79), and the popular `black` formatter default (88).

**Verification Code**

```python
def test_line_length(response: str, line_length: int = 79) -> bool:
    """
    Checks if the Python code adheres to a specific maximum line length
    (rule 'E501').
    """
    return _run_ruff_check(
        response,
        "--select", "E501",
        "--line-length", str(line_length)
    )
```

Figure 7: Full version of *style_3* instruction from VERICODE taxonomy.

**ID: `logic_3`**  **Category:** Logic & Code Patterns

**Description**

Enforce strict limits on the number of branches within functions to reduce cyclomatic complexity and improve maintainability.

**Generation Prompt**

*Ensure each function or method has at most {max_branches} branches, where branches include `if`, `elif`, `else` statements, `for` loops, `try-except` clauses, and `match-case` statements.*

**Edit Prompt**

*Simplify code so that each function or method has at most {max_branches} branches, where branches include `if`, `elif`, `else` statements, `for` loops, `try-except` clauses, and `match-case` statements.*

**Parameters**

`max_branches: int = 2`

**Notes**

This instruction limits the total number of branches per function using Ruff's `PLR0912` rule. Recommended values for challenging snippet-level evaluation settings are 2-4, with 2 as the default.

**Verification Code**

```python
def test_max_branch(response: str, max_branches: int = 2) -> bool:
    """
    Checks for maximum branches per function.
    """
    return _run_ruff_check(
        response,
        "--select", "PLR0912",
        "--config", f"lint.pylint.max-branches={max_branches}"
    )
```

Figure 8: Full version of *logic_3* instruction from VERICODE taxonomy.

**ID: doc_3**    **Category:** Documentation & Commenting

**Description**

Ensure all docstrings comply with the specified convention (Google, NumPy, or PEP 257) for proper formatting, placement, and content.

**Generation Prompt**

*Document the code fully by including docstrings that follow the {convention}-style convention.*

**Edit Prompt**

*Update the code to be fully documented by adding missing docstrings and formatting all existing docstrings to follow the {convention}-style convention.*

**Parameters**

```
convention: str = "pep257"
```

**Notes**

This instruction enforces the pydocstyle ( `D` ) ruleset with a specific convention. Valid conventions are `"google"` , `"numpy"` , or `"pep257"` . Each convention has different requirements for docstring structure, sections, and formatting. Google-style uses Args/Returns sections, NumPy-style uses Parameters/Returns with underlines, and PEP 257 provides basic formatting rules. The default is `"pep257"` for standard Python conventions.

**Verification Code**

```python
def test_docstring_convention(response: str, convention: str = "pep257") -> bool:
    """
    Checks for docstrings following the specified convention.
    """
    return _run_ruff_check(
        response,
        "--select", "D",
        "--config", f"lint.pydocstyle.convention='{convention}'"
    )
```

Figure 9: Full version of *doc_3* instruction from VERICODE taxonomy.

**ID: error_3**        **Category:** Error Handling & Exception Management

**Description**

Modernize exception handling by replacing legacy OSError aliases with the idiomatic and future-proof OSError base exception.

**Generation Prompt**

*Use the canonical `OSError` exception instead of its aliases.*

**Edit Prompt**

*Replace all uses of `OSError` aliases with the canonical `OSError` exception itself.*

**Parameters**

None

**Notes**

This instruction enforces the pyupgrade ( `UP024` ) rule. It identifies uses of exception types that are aliases for the built-in `OSError` . The refactoring requires replacing these legacy aliases, such as `IOError` and `WindowsError` , with `OSError` in all `raise` and `except` statements to create more modern, future-proof code.

**Verification Code**

```python
def test_os_error_alias(response: str) -> bool:
    """
    Checks for uses of exceptions that alias OSError.
    """
    return _run_ruff_check(response, "--select", "UP024")
```

Figure 10: Full version of *error_3* instruction from VERICODE taxonomy.

**ID: `library_1`**     **Category:** Library & API Constraints

**Description**

Replace all legacy file system operations—including os and os.path functions, the built-in open(), and glob—with their modern pathlib equivalents.

**Generation Prompt**

*Use pathlib equivalents instead of functions from `os`, `os.path`, `glob`, and the built-in `open`. Wrap the resulting Path object with `str()` where the surrounding code requires a string path to maintain functionality.*

**Edit Prompt**

*Replace all functions from `os`, `os.path`, `glob`, and the built-in `open` with their pathlib equivalents. Wrap the resulting Path object with `str()` where the surrounding code requires a string path to maintain functionality.*

**Parameters**

None

**Notes**

This instruction enforces the complete flake8-use-pathlib ( `PTH` ) ruleset. It identifies functions from `os` , `os.path` , `glob` , and the builtin `open()` that have pathlib.Path equivalents and requires replacing them. To preserve unit test compatibility, operations that originally returned strings (like `os.path.dirname` ) should have their pathlib equivalents wrapped in `str()` . This maintains the same return types while modernizing the implementation.

Note: This may cause failures in test environments that mock `open()` but not `pathlib` , leading to a `FileNotFoundError` . Therefore, it is advisable to avoid this instruction for code snippets that use the built-in `open()` function.

**Verification Code**

```python
def test_use_pathlib(response: str) -> bool:
    """
    Checks that all file system operations use pathlib.
    """
    return _run_ruff_check(response, "--select", "PTH")
```

Figure 11: Full version of *library_1* instruction from VERICODE taxonomy.

## C  VIBE CHECKER TESTBED

### C.1  INSTRUCTION CATEGORY DISTRIBUTIONS

Figure 12 illustrates the complete distribution of instruction categories selected for both augmented benchmarks. As shown, the three most frequent categories are *Coding Logic*, *Coding Style*, and *Documentation*. The distributions also reflect the distinct focus of each benchmark: the algorithm-oriented LiveVibeBench features a higher proportion of *Coding Logic* instructions (42.3% vs. 35.9%), while the real-world-task-focused BigVibeBench includes more instructions related to *Error Management* and *Library Constraint* instructions (6.3% vs. 0.9% and 2.2% vs. 0.1% respectively).

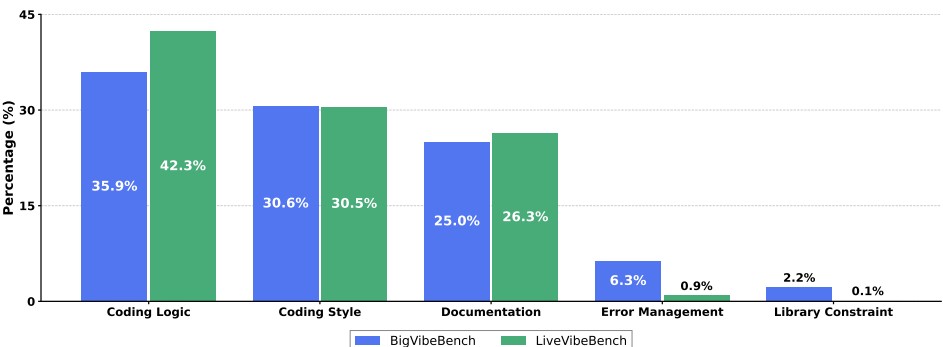

Figure 12: Percentage distribution of instruction categories on both augmented benchmarks.

### C.2  EVALUATION PROMPTS

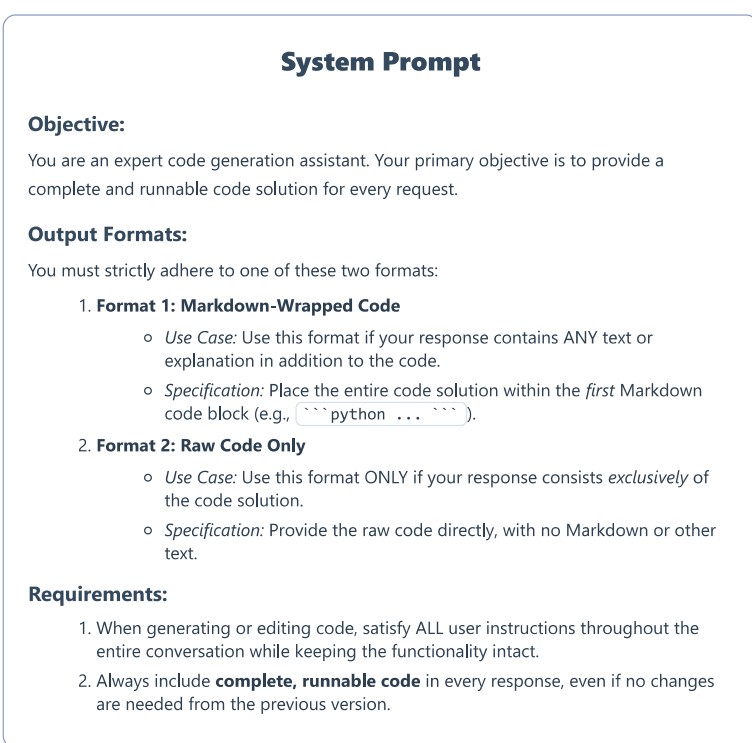

Figure 13: System prompt used for BigVibeBench. LiveVibeBench keeps the same wording with one minor change: "complete, runnable code" $\Rightarrow$ "complete Python functions," since algorithmic contest tasks often require only functions rather than full programs.

For BigVibeBench and LiveVibeBench, the system instruction and the evaluation prompts are shown in Figures 13 and 14. As we adopt BigCodeBench's original "instruct_prompt", we do not provide any additional evaluation prompt on this benchmark.

---

### Evaluation Prompts for LiveVibeBench

**( Task Type 1 ) — Standard Input/Output Tasks**

**Your Task:**

Write an executable Python function that solves the problem described in the prompt below.

**Requirements:**

- The function must read all necessary input from `stdin`.
- The function must print the final output to `stdout`.
- Simply call the function after the definition.

**Evaluation:**

We will evaluate your solution by running the code and comparing its standard output with the expected solution.

---

**( Task Type 2 ) — Functional Implementation Tasks**

**Your Task:**

Implement a function to solve the problem described in the prompt below.

**Requirements:**

- The primary function, which takes arguments as input and returns the final result, **must be named** `solve()`.
- The function **must NOT read from standard input** (e.g., using `input()` or `sys.stdin`). All required data will be passed in as function arguments.
- Any helper functions are permitted.
- Your code must only contain function definitions. **Strictly do not include a call to** `solve()`.

**Evaluation:**

We will evaluate your solution by first executing your code to load the function definitions, and then calling your `solve()` function directly with various test cases.

---

Figure 14: Evaluation prompts used in LiveVibeBench for the two task types.

# D  EXPERIMENTS

## D.1  DETAILS OF EVALUATED MODELS

For completeness and reproducibility, we list the comprehensive details of the 31 LLMs evaluated in our study, including their specific LMArena designations and the Elo ratings (Sep. 18, 2025) used for our human preference correlation analysis in Table 4.

Notably, on the LiveVibeBench benchmark, models demonstrate a significantly higher rate of failure to generate complete responses. These failures are attributed to either OpenRouter provider errors or exceeding the 32,768-token limit. In our experiments, each task is attempted up to three times, and a persistent failure is recorded as an error. To ensure the reliability of our results, we exclude models with an error rate exceeding 10%. Consequently, the LiveVibeBench analysis is conducted on the remaining 24 LLMs, with full results presented in Tables 6, 9, 10, and 12.

| Model | LMArena Name | Elo Rating | |
|---|---|---|---|
| | | w/o SC | w/ SC |
| Gemini 2.5 Pro | gemini-2.5-pro | **1468** | **1470** |
| Gemini 2.5 Flash | gemini-2.5-flash | 1422 | 1419 |
| Gemini 2.0 Flash | – | – | – |
| Gemini 2.0 Flash Lite | gemini-2.0-flash-lite-preview-02-05 | 1336 | 1352 |
| Claude 4 Opus | claude-opus-4-20250514-thinking-16k | 1430 | 1481 |
| Claude 4 Sonnet | claude-sonnet-4-20250514-thinking-32k | 1407 | 1460 |
| Claude 3.7 Sonnet | claude-3-7-sonnet-20250219-thinking-32k | 1353 | 1430 |
| Claude 3.5 Sonnet | Claude 3.5 Sonnet (10/22) | 1337 | 1418 |
| Claude 3.5 Haiku | claude-3-5-haiku-20241022 | 1285 | 1370 |
| Claude 3 Haiku | claude-3-haiku-20240307 | 1202 | 1287 |
| DeepSeek R1 0528 | deepseek-r1-0528 | 1436 | 1458 |
| DeepSeek V3 0324 | deepseek-v3-0324 | 1389 | 1431 |
| GPT 5 | gpt-5-high | 1440 | 1467 |
| o4 mini | o4-mini-2025-04-16 | 1380 | 1428 |
| o3 mini high | o3-mini-high | 1379 | 1421 |
| GPT 4.1 | gpt-4.1-2025-04-14 | 1399 | 1447 |
| GPT 4.1 mini | gpt-4.1-mini-2025-04-14 | 1371 | 1423 |
| GPT 4o | GPT-4o (08/06) | 1289 | 1352 |
| GPT 4o mini | GPT-4o-mini (07/18) | 1297 | 1340 |
| Grok 4 | grok-4-0709 | 1431 | 1440 |
| Grok 3 mini beta | grok-3-mini-beta | 1375 | 1384 |
| Qwen 3 235B A22B | qwen3-235b-a22b | 1392 | 1423 |
| Qwen 3 32B | qwen3-32b | 1375 | 1407 |
| Qwen 3 30B A3B | qwen3-30b-a3b | 1346 | 1378 |
| Qwen 2.5 72B Instruct | qwen2.5-72b-instruct | 1298 | 1346 |
| Qwen 2.5 Coder | qwen2.5-coder-32b-instruct | 1274 | 1325 |
| Gemma 3 27B | gemma-3-27b-it | 1348 | 1370 |
| Gemma 3 12B | gemma-3-12b-it | 1309 | 1332 |
| Mistral Medium 3 | mistral-medium-2505 | 1386 | 1421 |
| MiniMax M1 | minimax-m1 | 1368 | 1409 |
| Kimi K2 | kimi-k2-0711-preview | 1391 | 1454 |

Table 4: Details of the 31 LLMs evaluated in our experiments. For each model, we list its name as reported in this paper, its LMArena designation, and the Elo ratings used to analyze correlations with human preference. These ratings are from the September 18, 2025 leaderboard, presented under two conditions: with Style Control (w/ SC) and without (w/o SC).

## D.2 DETAILED RESULTS FOR FUNCTIONALITY

We present the detailed results for functionality on both benchmarks in Tables 5 and 6.

| Models | Base | Single-Turn Generation ↓ | | | | | Multi-Turn Editing ↓ | | | | |
|---|---|---|---|---|---|---|---|---|---|---|---|
| | | 1 Inst | 2 Inst | 3 Inst | 4 Inst | 5 Inst | 1 Inst | 2 Inst | 3 Inst | 4 Inst | 5 Inst |
| Gemini 2.5 Pro | 50.35 | 0.34 | 2.60 | 0.87 | -0.36 | 1.39 | 1.75 | 2.44 | 4.01 | 4.89 | 5.04 |
| Gemini 2.5 Flash | 47.37 | 0.74 | 1.12 | 2.60 | 1.31 | 2.41 | 0.93 | 1.12 | **1.48** | **2.98** | **3.72** |
| Gemini 2.0 Flash | 48.42 | 2.54 | 0.35 | 1.63 | 3.61 | 4.89 | 2.89 | 5.08 | 6.53 | 8.32 | 9.42 |
| Gemini 2.0 Flash Lite | 46.93 | 5.05 | 7.29 | 7.10 | 6.93 | 8.42 | 2.98 | 4.67 | 5.24 | 8.61 | 8.78 |
| Claude 4 Opus | 51.05 | -0.86 | **-2.23** | **-4.31** | **-1.72** | **-2.08** | 0.51 | **1.02** | 2.06 | 3.25 | 3.78 |
| Claude 4 Sonnet | 51.84 | -0.17 | -0.52 | 0.33 | 0.50 | 0.50 | 0.85 | 2.03 | 3.55 | 4.05 | 5.40 |
| Claude 3.7 Sonnet | 51.32 | 1.54 | 1.03 | 1.71 | 2.22 | 2.92 | 1.03 | 1.38 | 3.08 | 4.25 | 5.30 |
| Claude 3.5 Sonnet | 48.42 | 5.08 | 5.62 | 5.43 | 5.43 | 8.16 | 5.08 | 6.69 | 8.69 | 10.86 | 12.87 |
| Claude 3.5 Haiku | 46.58 | 5.28 | 4.34 | 6.98 | 7.34 | 9.42 | 6.98 | 9.98 | 15.44 | 17.13 | 21.28 |
| Claude 3 Haiku | 38.07 | 0.24 | 1.16 | 7.38 | 7.14 | 7.38 | 6.67 | 10.82 | 13.61 | 17.28 | 17.97 |
| DeepSeek R1 0528 | 49.21 | 1.24 | 0.18 | -1.24 | 1.61 | 3.03 | 1.61 | 1.08 | 3.92 | 3.03 | 4.27 |
| DeepSeek V3 0324 | 50.18 | 1.93 | 0.88 | 2.99 | 4.90 | 2.99 | 5.08 | 7.87 | 9.27 | 11.90 | 16.26 |
| GPT 5 | 46.49 | 0.56 | 5.66 | 2.26 | 3.20 | 1.89 | 1.70 | 2.82 | 4.35 | 5.27 | 5.46 |
| o4 mini | 52.28 | 4.02 | 9.39 | 5.87 | 7.38 | 9.56 | 2.18 | 4.71 | 7.04 | 7.04 | 8.05 |
| o3 mini high | 49.91 | 4.57 | 10.02 | 9.84 | 14.93 | 13.34 | 2.62 | 5.79 | 7.19 | 9.48 | 10.20 |
| GPT 4.1 | 47.54 | **-1.85** | -0.19 | 1.28 | 4.80 | 6.63 | 2.40 | 5.53 | 7.19 | 7.36 | 7.93 |
| GPT 4.1 mini | 49.04 | 0.55 | 1.61 | 2.69 | 4.49 | 5.38 | 4.30 | 6.44 | 6.99 | 8.24 | 8.77 |
| GPT 4o | 49.82 | 1.22 | 2.99 | 4.40 | 3.87 | 3.33 | 2.45 | 4.58 | 6.50 | 7.03 | 7.91 |
| GPT 4o mini | 46.05 | 5.91 | 5.52 | 7.23 | 6.28 | 7.99 | 7.80 | 6.47 | 9.71 | 11.62 | 11.62 |
| Grok 4 | **53.07** | 0.17 | 1.15 | 1.49 | 3.64 | 1.00 | 1.32 | 2.15 | 1.98 | 3.30 | 4.47 |
| Grok 3 mini beta | 48.77 | 2.52 | 4.86 | 7.91 | 8.10 | 9.35 | 2.15 | 4.86 | 5.76 | 7.73 | 9.17 |
| Qwen 3 235B A22B | 48.86 | 1.25 | 1.99 | 1.80 | 3.42 | 3.05 | 1.08 | 3.95 | 5.94 | 8.27 | 8.80 |
| Qwen 3 32B | 47.63 | 0.36 | 2.58 | 4.60 | 5.14 | 6.99 | 2.94 | 4.41 | 6.99 | 9.03 | 10.69 |
| Qwen 3 30B A3B | 46.40 | 2.63 | 3.58 | 3.41 | 5.09 | 7.18 | 1.87 | 4.91 | 5.86 | 7.56 | 7.93 |
| Qwen 2.5 72B Instruct | 44.39 | 6.53 | 8.52 | 10.88 | 11.08 | 12.05 | 8.90 | 10.88 | 12.26 | 14.24 | 16.02 |
| Qwen 2.5 Coder | 49.39 | 5.87 | 3.20 | 6.22 | 11.56 | 11.91 | 5.87 | 8.89 | 12.43 | 12.98 | 12.98 |
| Gemma 3 27B | 45.70 | 6.72 | 7.86 | 6.91 | 9.58 | 8.05 | 3.63 | 5.36 | 6.72 | 8.64 | 11.71 |
| Gemma 3 12B | 40.00 | 5.27 | 3.73 | 7.45 | 9.65 | 7.90 | 5.47 | 6.58 | 9.65 | 12.27 | 15.35 |
| Mistral Medium 3 | 45.44 | 5.22 | 8.30 | 9.07 | 9.46 | 10.81 | 5.79 | 6.18 | 8.69 | 9.07 | 9.86 |
| MiniMax M1 | 48.68 | 4.85 | 4.68 | 3.41 | 5.22 | 3.59 | **0.35** | 1.97 | 3.78 | 5.40 | 6.47 |
| Kimi K2 | 47.19 | -1.12 | -0.19 | -0.93 | 0.17 | 2.03 | 2.23 | 4.09 | 2.78 | 4.45 | 6.12 |

Table 5: Results for functionality on **BigVibeBench**. *Base* is the pass@1 score for the original query. All other cells report the functional regression rate (%) relative to the base. Lower is better, and negative values indicate improvement. Here, *k Inst* denotes the number of added instructions.

| Models | Base | Single-Turn Generation ↓ | | | | | Multi-Turn Editing ↓ | | | | |
|---|---|---|---|---|---|---|---|---|---|---|---|
| | | 1 Inst | 2 Inst | 3 Inst | 4 Inst | 5 Inst | 1 Inst | 2 Inst | 3 Inst | 4 Inst | 5 Inst |
| Gemini 2.5 Pro | **85.31** | -0.11 | 3.45 | 2.45 | 2.45 | 2.45 | 0.67 | 1.34 | 1.01 | 1.89 | 2.23 |
| Gemini 2.5 Flash | 74.50 | 3.56 | 5.34 | 8.01 | 5.60 | 6.74 | 0.12 | 1.14 | 1.65 | 3.44 | 3.69 |
| Gemini 2.0 Flash | 41.33 | 0.92 | 1.38 | 0.70 | 1.62 | 3.44 | 0.00 | 1.62 | 3.00 | 2.76 | 4.36 |
| Gemini 2.0 Flash Lite | 34.12 | 1.11 | -7.50 | -8.06 | -10.58 | -6.95 | 2.25 | 5.88 | 8.95 | 10.93 | 13.18 |
| Claude 4 Opus | 68.72 | 4.55 | 8.56 | 8.41 | 8.13 | 8.96 | 2.07 | 1.38 | 1.51 | 2.34 | 2.34 |
| Claude 4 Sonnet | 66.35 | 4.57 | 5.00 | 3.71 | 6.99 | 9.00 | 0.42 | 0.86 | 1.15 | 1.72 | 2.14 |
| Claude 3.7 Sonnet | 61.80 | -0.31 | 2.30 | 3.37 | 1.68 | 4.90 | **-0.47** | **0.45** | **0.92** | **1.23** | **1.99** |
| Claude 3.5 Sonnet | 45.40 | 1.67 | 2.49 | 2.09 | 5.22 | 6.48 | 2.09 | 5.22 | 7.09 | 8.06 | 11.70 |
| Claude 3.5 Haiku | 37.63 | 1.51 | 5.53 | 9.06 | 7.81 | 11.08 | 6.54 | 14.86 | 17.88 | 19.90 | 23.92 |
| Claude 3 Haiku | 22.09 | 11.18 | 19.74 | 21.05 | 25.35 | 30.06 | 6.02 | 8.60 | 12.04 | 13.31 | 16.34 |
| DeepSeek V3 0324 | 57.25 | 1.15 | 4.30 | 6.29 | 7.11 | 6.95 | 1.48 | 6.95 | 7.62 | 13.57 | 17.55 |
| GPT 5 | 71.47 | 1.72 | 2.13 | 3.32 | 7.16 | 6.76 | 2.25 | 4.24 | 5.57 | 7.43 | 9.02 |
| o4 mini | 80.95 | 5.74 | 9.02 | 9.02 | 11.37 | 12.29 | 3.63 | 8.91 | 10.19 | 11.71 | 15.92 |
| GPT 4.1 | 53.08 | -2.86 | -1.60 | 1.60 | 2.15 | 3.75 | 1.07 | 5.18 | 6.25 | 6.78 | 9.29 |
| GPT 4.1 mini | 58.86 | 3.53 | 7.88 | 8.85 | 9.97 | 10.79 | 1.44 | 4.99 | 7.73 | 8.21 | 8.85 |
| GPT 4o | 42.75 | 0.23 | 0.23 | 4.00 | 2.67 | 1.54 | 1.78 | 5.10 | 8.65 | 8.42 | 9.75 |
| GPT 4o mini | 22.27 | **-11.50** | **-11.50** | **-18.77** | **-15.36** | **-12.80** | 2.51 | 9.34 | 8.94 | 10.60 | 12.75 |
| Grok 3 mini beta | 65.97 | 2.58 | 3.15 | 6.75 | 12.93 | 11.93 | 0.86 | 3.30 | 5.46 | 6.03 | 7.90 |
| Qwen 3 30B A3B | 72.42 | 0.26 | 0.66 | 1.05 | 0.40 | 1.19 | 0.52 | 1.96 | 1.84 | 3.53 | 4.20 |
| Qwen 2.5 72B Instruct | 39.05 | 0.97 | 1.95 | 4.84 | 5.33 | 8.02 | 3.87 | 6.79 | 9.22 | 8.96 | 10.68 |
| Gemma 3 27B | 35.92 | 3.42 | 3.67 | 5.01 | 5.01 | 9.49 | 1.03 | 6.32 | 10.27 | 8.16 | 12.67 |
| Gemma 3 12B | 29.29 | 2.90 | -4.85 | -1.60 | -2.90 | 1.30 | 4.54 | 10.99 | 15.23 | 14.24 | 18.44 |
| Mistral Medium 3 | 40.66 | 3.03 | 7.45 | 6.98 | 3.71 | 4.89 | -0.25 | 2.78 | 2.78 | 4.65 | 5.36 |
| Kimi K2 | 63.58 | 8.92 | 15.48 | 16.07 | 15.48 | 16.36 | 2.64 | 5.63 | 9.50 | 12.49 | 12.79 |

Table 6: Results for functionality on **LiveVibeBench**. *Base* is the pass@1 score for the original query. All other cells report the functional regression rate (%) relative to the base. Lower is better, and negative values indicate improvement. Here, *k Inst* denotes the number of added instructions.

## D.3 Detailed Results for Instruction Following

The detailed results for instruction-level and task-level IF scores on both benchmarks are provided in Tables 7, 8, 9, and 10.

| Models | Single-Turn Generation ↑ | | | | | Multi-Turn Editing ↑ | | | | |
|---|---|---|---|---|---|---|---|---|---|---|
| | 1 Inst | 2 Inst | 3 Inst | 4 Inst | 5 Inst | 1 Inst | 2 Inst | 3 Inst | 4 Inst | 5 Inst |
| Gemini 2.5 Pro | 82.19 | 78.03 | 79.18 | 78.82 | 79.47 | 84.56 | 82.54 | 81.73 | 81.54 | 80.47 |
| Gemini 2.5 Flash | 81.67 | 77.81 | 77.34 | 75.35 | 75.91 | 78.68 | 75.35 | 74.62 | 73.57 | 73.25 |
| Gemini 2.0 Flash | 73.42 | 72.76 | 72.95 | 72.35 | 72.04 | 78.86 | 75.39 | 74.65 | 73.95 | 73.30 |
| Gemini 2.0 Flash Lite | 70.44 | 69.96 | 69.30 | 68.62 | 68.89 | 74.39 | 71.01 | 70.58 | 69.39 | 68.82 |
| Claude 4 Opus | **88.77** | **87.46** | **86.05** | **85.55** | **85.60** | 87.02 | 85.75 | 85.18 | 84.87 | 84.30 |
| Claude 4 Sonnet | 84.91 | 82.54 | 81.37 | 81.29 | 81.37 | 86.40 | 85.39 | 84.85 | 84.12 | 83.98 |
| Claude 3.7 Sonnet | 80.26 | 76.27 | 75.47 | 74.25 | 74.26 | 81.58 | 79.82 | 78.83 | 78.62 | 78.18 |
| Claude 3.5 Sonnet | 80.61 | 77.54 | 76.02 | 75.18 | 74.70 | 84.21 | 80.53 | 79.39 | 78.49 | 77.49 |
| Claude 3.5 Haiku | 64.56 | 64.74 | 63.71 | 63.07 | 63.82 | 79.91 | 73.38 | 71.26 | 68.20 | 65.60 |
| Claude 3 Haiku | 67.89 | 65.09 | 64.88 | 63.73 | 64.53 | 76.32 | 72.63 | 72.11 | 70.96 | 70.56 |
| DeepSeek R1 0528 | 74.04 | 69.78 | 69.01 | 69.28 | 67.71 | 77.02 | 74.12 | 72.37 | 71.69 | 71.05 |
| DeepSeek V3 0324 | 67.89 | 63.77 | 64.04 | 63.88 | 65.09 | 73.95 | 70.61 | 70.41 | 69.23 | 67.72 |
| GPT 5 | 82.89 | 82.28 | 81.96 | 81.64 | 81.77 | 84.91 | 85.18 | 85.94 | 85.83 | **86.39** |
| o4 mini | 84.82 | 84.21 | 83.25 | 83.68 | 84.25 | 88.51 | 86.10 | 84.18 | 83.60 | 83.28 |
| o3 mini high | 80.70 | 75.79 | 73.63 | 72.79 | 71.68 | 82.46 | 80.31 | 79.71 | 78.38 | 78.16 |
| GPT 4.1 | 81.40 | 78.07 | 78.60 | 77.28 | 77.75 | 82.63 | 80.31 | 79.09 | 78.36 | 77.79 |
| GPT 4.1 mini | 78.16 | 75.57 | 75.15 | 74.19 | 73.42 | 79.21 | 76.71 | 75.88 | 74.08 | 73.09 |
| GPT 4o | 77.46 | 74.87 | 74.56 | 73.25 | 73.44 | 85.09 | 82.37 | 80.79 | 79.45 | 78.35 |
| GPT 4o mini | 76.40 | 73.82 | 73.13 | 73.33 | 73.32 | 78.16 | 76.40 | 75.18 | 74.10 | 73.60 |
| Grok 4 | 87.11 | 85.61 | 84.77 | 84.39 | 84.81 | 88.51 | 87.19 | **86.93** | **86.29** | 85.37 |
| Grok 3 mini beta | 82.81 | 80.04 | 78.25 | 77.81 | 77.46 | 79.21 | 77.46 | 76.49 | 75.37 | 75.05 |
| Qwen 3 235B A22B | 83.95 | 81.05 | 80.38 | 79.30 | 78.63 | 85.09 | 81.89 | 80.50 | 80.13 | 78.89 |
| Qwen 3 32B | 76.75 | 72.81 | 71.81 | 70.92 | 71.35 | 82.02 | 80.53 | 78.92 | 77.39 | 76.33 |
| Qwen 3 30B A3B | 73.42 | 71.67 | 70.79 | 69.43 | 69.81 | 79.91 | 78.51 | 78.07 | 77.17 | 76.54 |
| Qwen 2.5 72B Instruct | 73.68 | 72.32 | 71.78 | 70.48 | 70.33 | 79.47 | 75.57 | 75.32 | 73.88 | 72.75 |
| Qwen 2.5 Coder | 71.40 | 67.11 | 67.57 | 65.42 | 65.77 | 73.33 | 71.71 | 71.26 | 69.76 | 68.86 |
| Gemma 3 27B | 68.42 | 67.06 | 65.50 | 64.56 | 65.02 | 73.60 | 69.65 | 69.30 | 68.14 | 66.72 |
| Gemma 3 12B | 65.96 | 66.14 | 65.44 | 65.26 | 65.00 | 67.54 | 67.76 | 67.19 | 66.05 | 64.95 |
| Mistral Medium 3 | 73.60 | 72.11 | 71.02 | 70.79 | 70.54 | 76.05 | 75.44 | 74.06 | 73.11 | 71.65 |
| MiniMax M1 | 74.12 | 70.75 | 71.70 | 71.07 | 70.89 | 77.63 | 74.30 | 74.06 | 73.60 | 72.95 |
| Kimi K2 | 85.00 | 83.46 | 81.46 | 80.42 | 79.14 | **89.12** | **87.46** | 86.70 | 85.09 | 84.19 |

Table 7: Instruction-level IF scores on **BigVibeBench**. Higher is better.

| Models | Single-Turn Generation ↑ | | | | | Multi-Turn Editing ↑ | | | | |
|---|---|---|---|---|---|---|---|---|---|---|
| | 1 Inst | 2 Inst | 3 Inst | 4 Inst | 5 Inst | 1 Inst | 2 Inst | 3 Inst | 4 Inst | 5 Inst |
| Gemini 2.5 Pro | 82.19 | 60.70 | 48.16 | 37.46 | 30.70 | 84.56 | 68.33 | 55.61 | 44.21 | 33.68 |
| Gemini 2.5 Flash | 81.67 | 61.05 | 43.68 | 30.53 | 25.70 | 78.68 | 56.75 | 40.96 | 29.12 | 21.75 |
| Gemini 2.0 Flash | 73.42 | 53.77 | 39.47 | 26.40 | 18.16 | 78.86 | 59.39 | 44.56 | 32.46 | 22.46 |
| Gemini 2.0 Flash Lite | 70.44 | 48.60 | 32.63 | 22.02 | 15.26 | 74.39 | 50.61 | 35.18 | 24.12 | 15.35 |
| Claude 4 Opus | **88.77** | **76.32** | **64.21** | **52.98** | **46.75** | 87.02 | 73.16 | 61.05 | 51.32 | 42.11 |
| Claude 4 Sonnet | 84.91 | 67.19 | 52.28 | 42.98 | 35.26 | 86.40 | 72.54 | 61.23 | 51.05 | 42.89 |
| Claude 3.7 Sonnet | 80.26 | 56.93 | 39.91 | 27.46 | 22.28 | 81.58 | 63.51 | 48.51 | 38.16 | 29.39 |
| Claude 3.5 Sonnet | 80.61 | 59.74 | 42.98 | 32.37 | 24.47 | 84.21 | 66.40 | 52.54 | 42.02 | 32.28 |
| Claude 3.5 Haiku | 64.56 | 42.46 | 26.14 | 15.53 | 10.09 | 79.91 | 57.63 | 42.72 | 30.00 | 19.82 |
| Claude 3 Haiku | 67.89 | 41.84 | 26.05 | 16.93 | 11.93 | 76.32 | 53.60 | 37.89 | 26.49 | 18.77 |
| DeepSeek R1 0528 | 74.04 | 49.21 | 33.42 | 25.00 | 17.63 | 77.02 | 55.18 | 38.16 | 26.67 | 18.51 |
| DeepSeek V3 0324 | 67.89 | 39.21 | 24.74 | 15.00 | 10.88 | 73.95 | 52.02 | 37.19 | 24.65 | 14.74 |
| GPT 5 | 82.89 | 67.63 | 54.04 | 42.98 | 34.39 | 84.91 | 72.37 | 62.98 | 55.26 | **48.51** |
| o4 mini | 84.82 | 70.79 | 57.11 | 47.98 | 41.32 | 88.51 | 74.74 | 61.23 | 50.09 | 41.84 |
| o3 mini high | 80.70 | 60.61 | 45.88 | 36.40 | 28.25 | 82.46 | 66.32 | 53.16 | 42.11 | 34.56 |
| GPT 4.1 | 81.40 | 59.91 | 47.81 | 35.44 | 28.16 | 82.63 | 65.26 | 50.88 | 39.82 | 31.58 |
| GPT 4.1 mini | 78.16 | 56.23 | 41.49 | 30.26 | 21.75 | 79.21 | 59.39 | 44.74 | 33.68 | 25.53 |
| GPT 4o | 77.46 | 55.00 | 39.56 | 27.63 | 20.79 | 85.09 | 68.33 | 52.72 | 40.88 | 30.70 |
| GPT 4o mini | 76.40 | 53.86 | 38.68 | 29.30 | 21.84 | 78.16 | 59.74 | 44.12 | 32.54 | 23.42 |
| Grok 4 | 87.11 | 73.42 | 60.18 | 51.84 | 43.16 | 88.51 | 76.40 | 66.05 | **55.96** | 47.19 |
| Grok 3 mini beta | 82.81 | 64.21 | 48.86 | 36.58 | 28.42 | 79.21 | 61.40 | 46.93 | 34.91 | 25.96 |
| Qwen 3 235B A22B | 83.95 | 66.75 | 52.28 | 42.28 | 31.93 | 85.09 | 67.63 | 51.84 | 41.32 | 32.28 |
| Qwen 3 32B | 76.75 | 53.86 | 36.49 | 26.58 | 20.70 | 82.02 | 65.79 | 51.49 | 39.82 | 30.70 |
| Qwen 3 30B A3B | 73.42 | 52.46 | 36.23 | 25.79 | 19.56 | 79.91 | 62.46 | 48.16 | 37.46 | 29.56 |
| Qwen 2.5 72B Instruct | 73.68 | 53.07 | 37.37 | 24.56 | 16.84 | 79.47 | 60.53 | 45.70 | 33.25 | 24.21 |
| Qwen 2.5 Coder | 71.40 | 44.82 | 30.70 | 20.09 | 12.81 | 73.33 | 52.46 | 36.93 | 24.04 | 15.88 |
| Gemma 3 27B | 68.42 | 44.56 | 27.11 | 16.93 | 10.96 | 73.60 | 48.42 | 33.33 | 21.93 | 14.12 |
| Gemma 3 12B | 65.96 | 44.39 | 27.98 | 18.42 | 11.05 | 67.54 | 46.75 | 31.58 | 20.09 | 12.81 |
| Mistral Medium 3 | 73.60 | 51.93 | 36.32 | 25.09 | 16.05 | 76.05 | 58.33 | 41.58 | 28.60 | 19.30 |
| MiniMax M1 | 74.12 | 51.23 | 37.98 | 28.07 | 20.35 | 77.63 | 57.19 | 42.11 | 31.75 | 22.98 |
| Kimi K2 | 85.00 | 68.86 | 53.68 | 41.23 | 30.18 | **89.12** | **77.11** | **66.40** | 53.95 | 44.04 |

Table 8: Task-level IF scores on **BigVibeBench**. Higher scores are better.

| Models | Single-Turn Generation ↑ | | | | | Multi-Turn Editing ↑ | | | | |
|---|---|---|---|---|---|---|---|---|---|---|
| | 1 Inst | 2 Inst | 3 Inst | 4 Inst | 5 Inst | 1 Inst | 2 Inst | 3 Inst | 4 Inst | 5 Inst |
| Gemini 2.5 Pro | 75.83 | 74.60 | 76.21 | 77.37 | 76.87 | 78.96 | 77.87 | 78.99 | 78.89 | 78.98 |
| Gemini 2.5 Flash | 66.54 | 67.11 | 67.84 | 68.01 | 68.13 | 72.80 | 71.42 | 69.64 | 68.93 | 67.89 |
| Gemini 2.0 Flash | 61.71 | 61.37 | 61.42 | 62.01 | 62.48 | 74.41 | 71.37 | 70.36 | 69.45 | 69.52 |
| Gemini 2.0 Flash Lite | 62.94 | 63.93 | 65.28 | 65.40 | 65.69 | 67.30 | 64.36 | 63.06 | 61.37 | 61.19 |
| Claude 4 Opus | 78.86 | 76.02 | 77.54 | 77.65 | 78.75 | **85.59** | 85.36 | 85.21 | 84.31 | 84.11 |
| Claude 4 Sonnet | 75.73 | 74.69 | 75.29 | 74.27 | 75.20 | 84.45 | 85.40 | 85.24 | 84.50 | 83.87 |
| Claude 3.7 Sonnet | 72.42 | 68.53 | 68.18 | 68.06 | 68.38 | 79.53 | 78.48 | 77.91 | 77.18 | 76.76 |
| Claude 3.5 Sonnet | 70.52 | 68.25 | 68.56 | 68.15 | 67.28 | 80.57 | 76.40 | 75.23 | 74.29 | 73.12 |
| Claude 3.5 Haiku | 63.22 | 60.19 | 61.45 | 61.80 | 62.77 | 78.67 | 75.50 | 72.51 | 70.52 | 68.99 |
| Claude 3 Haiku | 61.61 | 59.91 | 60.98 | 60.97 | 60.45 | 72.80 | 71.28 | 69.23 | 67.39 | 67.45 |
| DeepSeek V3 0324 | 52.80 | 54.74 | 55.67 | 55.81 | 55.79 | 70.05 | 66.49 | 65.72 | 64.83 | 62.71 |
| GPT 5 | **82.18** | **82.18** | **81.86** | **82.09** | **82.82** | **85.59** | **86.30** | **87.05** | **85.85** | **85.76** |
| o4 mini | 73.18 | 72.27 | 73.33 | 73.08 | 73.82 | 81.52 | 80.47 | 79.53 | 76.99 | 75.81 |
| GPT 4.1 | 68.63 | 65.12 | 66.76 | 66.30 | 66.67 | 74.12 | 72.89 | 71.97 | 71.75 | 70.81 |
| GPT 4.1 mini | 67.20 | 66.40 | 68.63 | 67.89 | 67.41 | 71.75 | 69.72 | 69.23 | 68.53 | 67.72 |
| GPT 4o | 60.85 | 60.85 | 61.48 | 61.75 | 61.93 | 76.40 | 73.13 | 71.94 | 70.31 | 69.93 |
| GPT 4o mini | 65.88 | 65.40 | 65.72 | 65.64 | 65.63 | 73.93 | 71.85 | 70.36 | 68.39 | 67.22 |
| Grok 3 mini beta | 70.05 | 70.05 | 69.61 | 69.12 | 68.99 | 78.67 | 75.45 | 73.46 | 72.23 | 71.09 |
| Qwen 3 30B A3B | 67.77 | 62.89 | 63.76 | 64.10 | 63.00 | 73.27 | 71.04 | 70.74 | 68.98 | 68.91 |
| Qwen 2.5 72B Instruct | 64.83 | 63.65 | 65.97 | 64.88 | 66.14 | 74.50 | 69.43 | 69.38 | 67.70 | 67.51 |
| Gemma 3 27B | 61.99 | 62.09 | 62.53 | 63.51 | 63.56 | 66.92 | 64.83 | 64.01 | 63.44 | 63.41 |
| Gemma 3 12B | 61.33 | 62.09 | 62.46 | 63.25 | 62.29 | 66.92 | 63.65 | 63.44 | 61.73 | 60.76 |
| Mistral Medium 3 | 62.37 | 61.28 | 61.90 | 62.44 | 62.45 | 69.67 | 64.31 | 63.95 | 63.67 | 63.37 |
| Kimi K2 | 62.75 | 63.65 | 64.80 | 64.38 | 64.76 | 76.97 | 74.64 | 74.31 | 72.94 | 73.65 |

Table 9: Instruction-level IF scores on **LiveVibeBench**. Higher is better.

| Models | Single-Turn Generation ↑ | | | | | Multi-Turn Editing ↑ | | | | |
|---|---|---|---|---|---|---|---|---|---|---|
| | 1 Inst | 2 Inst | 3 Inst | 4 Inst | 5 Inst | 1 Inst | 2 Inst | 3 Inst | 4 Inst | 5 Inst |
| Gemini 2.5 Pro | 75.83 | 56.78 | 45.50 | 37.63 | 29.57 | 78.96 | 61.61 | 51.18 | 41.04 | 32.80 |
| Gemini 2.5 Flash | 66.54 | 45.97 | 32.89 | 23.03 | 17.06 | 72.80 | 51.09 | 34.98 | 25.31 | 17.82 |
| Gemini 2.0 Flash | 61.71 | 37.25 | 22.18 | 13.46 | 8.44 | 74.41 | 51.28 | 36.02 | 24.64 | 17.73 |
| Gemini 2.0 Flash Lite | 62.94 | 41.33 | 27.68 | 18.39 | 12.89 | 67.30 | 42.75 | 27.11 | 17.35 | 10.62 |
| Claude 4 Opus | 78.86 | 57.91 | 47.96 | 38.96 | 35.17 | **85.59** | 72.89 | 61.71 | 52.04 | 43.70 |
| Claude 4 Sonnet | 75.73 | 56.40 | 44.17 | 35.36 | 28.53 | 84.45 | 73.46 | 62.37 | 52.70 | 44.64 |
| Claude 3.7 Sonnet | 72.42 | 47.01 | 31.85 | 23.51 | 18.96 | 79.53 | 62.46 | 48.53 | 38.58 | 30.33 |
| Claude 3.5 Sonnet | 70.52 | 47.01 | 31.94 | 22.37 | 14.88 | 80.57 | 60.28 | 44.45 | 35.73 | 27.20 |
| Claude 3.5 Haiku | 63.22 | 35.92 | 22.84 | 16.40 | 11.66 | 78.67 | 58.58 | 41.23 | 30.52 | 22.46 |
| Claude 3 Haiku | 61.61 | 36.68 | 23.13 | 15.83 | 9.95 | 72.80 | 52.32 | 36.59 | 26.82 | 18.58 |
| DeepSeek V3 0324 | 52.80 | 29.76 | 19.24 | 11.28 | 7.77 | 70.05 | 45.31 | 31.56 | 20.38 | 11.37 |
| GPT 5 | **82.18** | **68.53** | **55.17** | **47.01** | **40.95** | **85.59** | **74.50** | **66.64** | **57.35** | **50.14** |
| o4 mini | 73.18 | 53.93 | 43.22 | 33.36 | 27.20 | 81.52 | 66.64 | 54.60 | 42.84 | 32.61 |
| GPT 4.1 | 68.63 | 42.27 | 29.48 | 20.57 | 13.65 | 74.12 | 54.31 | 41.04 | 32.70 | 24.08 |
| GPT 4.1 mini | 67.20 | 43.60 | 30.71 | 21.99 | 14.88 | 71.75 | 49.67 | 34.60 | 26.26 | 18.20 |
| GPT 4o | 60.85 | 36.40 | 22.75 | 14.98 | 9.95 | 76.40 | 53.93 | 37.91 | 28.44 | 20.38 |
| GPT 4o mini | 65.88 | 42.65 | 28.25 | 20.19 | 14.41 | 73.93 | 52.32 | 36.21 | 24.93 | 17.06 |
| Grok 3 mini beta | 70.05 | 51.09 | 38.10 | 28.15 | 20.66 | 78.67 | 58.39 | 43.89 | 32.99 | 24.64 |
| Qwen 3 30B A3B | 67.77 | 42.09 | 29.67 | 22.65 | 14.50 | 73.27 | 52.89 | 39.72 | 29.10 | 21.90 |
| Qwen 2.5 72B Instruct | 64.83 | 40.76 | 28.63 | 18.86 | 15.92 | 74.50 | 50.24 | 37.06 | 26.73 | 19.24 |
| Gemma 3 27B | 61.99 | 37.73 | 24.17 | 16.30 | 11.00 | 66.92 | 41.71 | 27.30 | 17.91 | 12.32 |
| Gemma 3 12B | 61.33 | 38.20 | 24.17 | 16.30 | 9.19 | 66.92 | 41.71 | 26.54 | 16.40 | 10.05 |
| Mistral Medium 3 | 62.37 | 37.25 | 23.13 | 15.83 | 9.86 | 69.67 | 42.94 | 27.96 | 18.48 | 11.66 |
| Kimi K2 | 62.75 | 41.61 | 27.77 | 19.05 | 11.94 | 76.97 | 57.35 | 44.17 | 35.73 | 27.87 |

Table 10: Task-level IF scores on **LiveVibeBench**. Higher is better.

# E ANALYSIS

## E.1 INSTRUCTION POSITION ANALYSIS

As listed in Tables 11 and 12, we also provide detailed results for per-position instruction-level IF scores on both benchmarks.

| Models | Single-Turn Generation ↑ | | | | | Multi-Turn Editing ↑ | | | | |
|---|---|---|---|---|---|---|---|---|---|---|
| | Pos 1 | Pos 2 | Pos 3 | Pos 4 | Pos 5 | Pos 1 | Pos 2 | Pos 3 | Pos 4 | Pos 5 |
| Gemini 2.5 Pro | 81.40 | 79.91 | 78.86 | 78.60 | 78.60 | 79.91 | 78.16 | 79.56 | 81.58 | 83.16 |
| Gemini 2.5 Flash | 79.82 | 74.04 | 75.44 | 75.00 | 75.26 | 72.19 | 69.74 | 72.81 | 73.25 | 78.25 |
| Gemini 2.0 Flash | 74.56 | 71.23 | 72.02 | 70.96 | 71.40 | 72.81 | 71.75 | 74.47 | 73.77 | 73.68 |
| Gemini 2.0 Flash Lite | 72.11 | 68.77 | 67.89 | 67.89 | 67.81 | 68.16 | 67.81 | 69.12 | 68.25 | 70.79 |
| Claude 4 Opus | **87.46** | **85.18** | **85.18** | **83.95** | **86.23** | 84.56 | 82.81 | 84.82 | 83.60 | 85.70 |
| Claude 4 Sonnet | 83.77 | 79.65 | 80.26 | 81.67 | 81.49 | 84.21 | 80.18 | 83.95 | 84.39 | 87.19 |
| Claude 3.7 Sonnet | 77.81 | 73.16 | 72.19 | 73.07 | 75.09 | 77.63 | 76.75 | 76.93 | 78.60 | 80.88 |
| Claude 3.5 Sonnet | 77.46 | 72.81 | 72.72 | 75.18 | 75.35 | 76.40 | 75.53 | 75.09 | 78.33 | 82.11 |
| Claude 3.5 Haiku | 65.00 | 62.46 | 61.49 | 62.89 | 67.28 | 60.70 | 63.68 | 64.65 | 65.79 | 73.16 |
| Claude 3 Haiku | 67.89 | 62.72 | 62.28 | 63.86 | 65.88 | 72.11 | 66.40 | 69.47 | 68.95 | 75.88 |
| DeepSeek R1 0528 | 69.30 | 66.58 | 66.58 | 67.63 | 67.98 | 68.86 | 68.33 | 70.53 | 71.32 | 76.23 |
| DeepSeek V3 0324 | 67.63 | 62.81 | 64.47 | 64.74 | 65.79 | 61.67 | 63.95 | 68.25 | 69.12 | 75.61 |
| GPT 5 | 82.72 | 81.40 | 81.05 | 81.58 | 82.11 | **86.05** | **86.05** | **87.02** | **85.96** | 86.84 |
| o4 mini | 85.53 | 82.81 | 83.07 | 83.77 | 86.05 | 81.40 | 80.44 | 82.11 | 84.04 | **88.42** |
| o3 mini high | 73.25 | 72.19 | 69.04 | 71.84 | 72.11 | 77.46 | 74.65 | 77.19 | 78.95 | 82.54 |
| GPT 4.1 | 78.95 | 76.75 | 76.40 | 78.33 | 78.33 | 78.68 | 75.88 | 77.63 | 77.19 | 79.56 |
| GPT 4.1 mini | 75.00 | 72.63 | 72.63 | 71.49 | 75.35 | 75.26 | 70.26 | 71.84 | 72.02 | 76.05 |
| GPT 4o | 78.77 | 72.72 | 71.40 | 71.75 | 72.54 | 78.42 | 75.79 | 77.19 | 78.42 | 81.93 |
| GPT 4o mini | 76.14 | 71.32 | 71.84 | 72.81 | 74.47 | 71.58 | 71.84 | 72.89 | 73.51 | 78.16 |
| Grok 4 | 86.32 | 83.77 | 84.39 | 83.86 | 85.70 | 84.91 | 84.56 | 85.70 | 84.74 | 86.93 |
| Grok 3 mini beta | 79.56 | 76.40 | 76.40 | 75.44 | 79.47 | 73.07 | 72.89 | 72.89 | 75.26 | 81.14 |
| Qwen 3 235B A22B | 82.54 | 78.16 | 77.28 | 77.28 | 77.89 | 77.98 | 76.84 | 77.54 | 80.61 | 81.49 |
| Qwen 3 32B | 73.68 | 70.61 | 71.75 | 69.82 | 70.88 | 76.49 | 73.42 | 76.23 | 77.02 | 78.51 |
| Qwen 3 30B A3B | 71.67 | 67.81 | 68.60 | 68.95 | 72.02 | 77.19 | 74.39 | 77.02 | 75.53 | 78.60 |
| Qwen 2.5 72B Instruct | 73.07 | 71.05 | 69.47 | 68.33 | 69.74 | 73.07 | 70.88 | 72.89 | 72.72 | 74.21 |
| Qwen 2.5 Coder | 68.07 | 64.74 | 64.74 | 64.39 | 66.93 | 67.98 | 65.96 | 69.12 | 69.56 | 71.67 |
| Gemma 3 27B | 68.07 | 64.30 | 63.86 | 64.74 | 64.12 | 65.96 | 65.35 | 67.63 | 66.75 | 67.89 |
| Gemma 3 12B | 67.98 | 66.14 | 64.30 | 63.60 | 62.98 | 65.00 | 62.98 | 65.35 | 63.77 | 67.63 |
| Mistral Medium 3 | 74.65 | 69.82 | 68.42 | 70.79 | 69.04 | 70.88 | 68.86 | 69.74 | 71.75 | 77.02 |
| MiniMax M1 | 72.37 | 70.70 | 70.09 | 70.00 | 71.32 | 70.61 | 70.61 | 72.89 | 73.86 | 76.75 |
| Kimi K2 | 83.42 | 79.12 | 77.46 | 78.25 | 77.46 | 84.47 | 83.86 | 82.19 | 83.51 | 86.93 |

Table 11: Instruction-position analysis on **BigVibeBench**. We report instruction-level IF scores under the setting of five instructions, comparing positions 1–5 in each setting. In Single-Turn Generation, position $i$ denotes the $i$-th item in the numbered instruction list given to the model. In Multi-Turn Editing, position $i$ indicates the $i$-th instruction introduced as a separate turn.

| Models | Single-Turn Generation ↑ | | | | | Multi-Turn Editing ↑ | | | | |
|---|---|---|---|---|---|---|---|---|---|---|
| | Pos 1 | Pos 2 | Pos 3 | Pos 4 | Pos 5 | Pos 1 | Pos 2 | Pos 3 | Pos 4 | Pos 5 |
| Gemini 2.5 Pro | 76.78 | 75.83 | 77.54 | 76.97 | 77.25 | 76.49 | 76.59 | 78.77 | 80.57 | 82.46 |
| Gemini 2.5 Flash | 70.52 | 67.39 | 69.57 | 67.58 | 65.59 | 66.16 | 64.93 | 67.39 | 67.87 | 73.08 |
| Gemini 2.0 Flash | 63.41 | 60.95 | 63.79 | 61.33 | 62.94 | 69.10 | 67.87 | 68.53 | 69.76 | 72.32 |
| Gemini 2.0 Flash Lite | 67.39 | 65.21 | 65.97 | 63.60 | 66.26 | 60.38 | 60.09 | 61.42 | 60.19 | 63.89 |
| Claude 4 Opus | 79.91 | 77.91 | 78.29 | 77.06 | 80.57 | 84.83 | 84.64 | 83.41 | 82.75 | 84.93 |
| Claude 4 Sonnet | 75.64 | 75.45 | 75.26 | 74.03 | 75.64 | 81.90 | 85.31 | 84.17 | 82.75 | **85.21** |
| Claude 3.7 Sonnet | 70.81 | 65.97 | 69.38 | 67.01 | 68.72 | 76.30 | 76.40 | 76.02 | 75.73 | 79.34 |
| Claude 3.5 Sonnet | 67.87 | 67.01 | 67.77 | 65.78 | 67.96 | 72.32 | 71.94 | 71.94 | 72.51 | 76.87 |
| Claude 3.5 Haiku | 63.03 | 62.46 | 61.52 | 62.65 | 64.17 | 67.58 | 66.26 | 68.06 | 68.44 | 74.60 |
| Claude 3 Haiku | 62.65 | 57.82 | 60.66 | 60.38 | 60.76 | 68.63 | 66.35 | 68.25 | 64.74 | 69.29 |
| DeepSeek V3 0324 | 55.64 | 56.11 | 56.11 | 55.55 | 55.55 | 58.29 | 59.34 | 62.75 | 64.74 | 68.44 |
| GPT 5 | **83.51** | **82.37** | **83.70** | **81.33** | **83.22** | **86.26** | **86.92** | **86.07** | **84.55** | 85.02 |
| o4 mini | 75.17 | 74.31 | 74.41 | 71.56 | 73.65 | 73.46 | 74.31 | 76.11 | 75.55 | 79.62 |
| GPT 4.1 | 67.20 | 65.97 | 68.44 | 64.55 | 67.20 | 71.28 | 69.76 | 72.42 | 69.29 | 71.28 |
| GPT 4.1 mini | 68.82 | 67.49 | 69.95 | 63.79 | 67.01 | 67.30 | 66.54 | 69.38 | 65.97 | 69.38 |
| GPT 4o | 62.84 | 61.61 | 63.32 | 60.28 | 61.61 | 68.44 | 68.15 | 71.09 | 68.06 | 73.93 |
| GPT 4o mini | 67.01 | 64.83 | 66.26 | 64.45 | 65.59 | 67.30 | 65.97 | 67.49 | 64.83 | 70.52 |
| Grok 3 mini beta | 69.10 | 67.39 | 69.48 | 68.44 | 70.52 | 67.11 | 69.86 | 69.76 | 72.23 | 76.49 |
| Qwen 3 30B A3B | 62.94 | 63.03 | 64.36 | 62.65 | 61.99 | 66.73 | 68.06 | 69.86 | 69.19 | 70.71 |
| Qwen 2.5 72B Instruct | 66.54 | 64.08 | 68.25 | 65.88 | 65.97 | 66.35 | 65.88 | 68.82 | 67.20 | 69.29 |
| Gemma 3 27B | 65.97 | 62.56 | 64.17 | 63.41 | 61.71 | 62.27 | 61.80 | 62.65 | 63.13 | 67.20 |
| Gemma 3 12B | 64.36 | 62.37 | 63.13 | 60.76 | 60.85 | 60.76 | 58.67 | 60.57 | 58.96 | 64.83 |
| Mistral Medium 3 | 63.32 | 61.23 | 63.41 | 61.71 | 62.56 | 60.38 | 58.67 | 63.79 | 64.83 | 69.19 |
| Kimi K2 | 66.64 | 64.17 | 65.97 | 62.94 | 64.08 | 72.61 | 72.51 | 74.69 | 73.27 | 75.17 |

Table 12: Instruction-position analysis on **LiveVibeBench**. We report instruction-level IF scores under the setting of five instructions, comparing positions 1–5 in each setting. In Single-Turn Generation, position $i$ denotes the $i$-th item in the numbered instruction list given to the model. In Multi-Turn Editing, position $i$ indicates the $i$-th instruction introduced as a separate turn.

## E.2 CORRELATION ANALYSIS

(a) BigVibeBench w/ Style Control

(b) LiveVibeBench w/ Style Control

(c) BigVibeBench w/o Style Control

(d) LiveVibeBench w/o Style Control

Figure 15: Human preference aligns best with a mix of IF and functionality. We correlate LMArena coding Elo with a composite score $\alpha\,\text{IF}+(1-\alpha)\,\text{Func}$, where $\alpha \in [0, 1]$ is the weight on IF (x-axis). We also vary the correlation type and toggle the style-control function on/off. Nevertheless, the peak correlation (starred) consistently occurs at a mixture of the two metrics across all settings.

