# OpenReview forum: "Vibe Checker: Aligning Code Evaluation with Human Preference"
_ICLR.cc/2026/Conference — Submitted to ICLR 2026_

### Official Review · Reviewer_sume · 2025-10-30

**Soundness:** 3
**Presentation:** 3
**Contribution:** 2
**Rating:** 4
**Confidence:** 4

**Summary:**

This paper proposes ‘vibe checker’, aiming at evaluating the instruction following ability to represent human preferences besides the function correctness. They create a taxonomy of 30 coding instructions and their corresponding deterministic verifiers, which can be used to augment existing evaluation suites. The taxonomy covers five categories, including coding style & conventions, logic & code patterns, documentation & commenting, error handling & exception management, and library & API constraints. The curated instructions are mainly for Python. Through evaluation on 31 leading LLMs, the authors find that non-functional instructions can cause noticeable performance regressions in functional correctness, revealing gaps in models’ instruction-following ability.

**Strengths:**

- By using executable, deterministic verifiers, the proposed method grounds qualitative “vibe” in reproducible and measurable criteria, improving reliability and comparability of evaluations.
- The evaluation results show that it’s still challenging for advanced LLMs to follow non-functional instructions while maintaining functional correctness, offering a valuable insight for future model development.

**Weaknesses:**

- The benchmarks used in this paper, BigCodeBench and LiveCodeBench, are all function-level code generation tasks for single files. However, “vibe coding” instructions often become more practical and complex at the project or multi-file level, where style consistency, documentation practices, and dependency management interact. For project-level instruction following, if the code generation task involves multiple files, curating relevant and non-conflict instructions will also be challenging.
- While the authors mention “practice grounding” as a design principle, the process remains vaguely described. It lacks evidence and reference on how manual review by the author team ensures real-world relevance.
- The results could be made more actionable by breaking down which categories or instruction types most strongly correlate with functional regressions, and why. This would help identify specific model weaknesses and guide targeted improvements.

**Questions:**

During benchmark construction, the paper states that an LLM was used to select relevant and non-conflicting instructions. How was this LLM-based selector evaluated for accuracy and reliability?

---

> ### Author Response · Authors · 2025-11-24
> **Response to Reviewer sume (Part 1)**
>
> We thank the reviewer for recognizing the value of our deterministic verifiers in grounding qualitative "vibe" and for acknowledging the insights offered for future model development. We appreciate your constructive feedback on the benchmark scope and result analysis. Below, we address your concerns regarding the benchmark level and technical details.
>
> > **W1: The benchmarks used in this paper, BigCodeBench and LiveCodeBench, are all function-level code generation tasks for single files.**
>
> We acknowledge that our current evaluation focuses on function-level tasks. However, we argue that this setting serves as a necessary and representative foundation for the current stage of LLM evaluation for the following reasons.
>
> - **Difficulty and Foundational Capability:** Even at the function level, current state-of-the-art models struggle significantly. As our results show, under five instructions, the best-performing models achieve a task-level IF score of only roughly 40% to 50%. Mastering these atomic constraints at the function level is a prerequisite for success in more complex, multi-file contexts where dependency management and consistency add further difficulty.
> - **Alignment with Human Preference Data:** Our primary motivation is to bridge the gap between benchmark metrics and real-world human preference. The current gold standard for human preference data, LMArena, is dominated by snippet-level or single-file coding interactions rather than repository-level tasks. Therefore, evaluating on function-level benchmarks ensures our analysis aligns with the distribution of currently available human preference data.
> - **Methodological Extensibility:** It is also important to note that our plug-in methodology is not limited to the function level. Since our taxonomy derives from industrial standards (e.g., linters), verifiable instructions can be injected into any coding context. For example, in repository-level benchmarks like SWE-Bench, our verifiers can assess whether the generated patches adhere to non-functional constraints, such as specific docstring conventions or coding patterns.
>
> ---
>
> > **W2: While the authors mention “practice grounding” as a design principle, the process remains vaguely described. It lacks evidence and reference on how manual review by the author team ensures real-world relevance.**
>
> Thank you for pointing this out. We agree that the “practice grounding” principle deserves a more concrete description, and we clarify the evidence here.
>
> **Source of Real-World Relevance.** The real-world relevance of our taxonomy stems primarily from its source material rather than subjective author preference. We utilize Ruff, an industry-standard tool that aggregates over 800 rules from widely adopted Python linters, including Flake8 (style), isort, pydocstyle (documentation), and pyupgrade (syntax modernization) etc. These tools are integrated into millions of repositories and CI/CD pipelines globally. Therefore, the instructions in VeriCode are natively grounded in the collective consensus of the software engineering community.
>
> **Role of Manual Review.** The role of our manual review is not to invent new "vibes," but to strictly curate this vast pool into a representative benchmark. Specifically, we focus on filtering out niche rules (e.g., specific framework-dependent constraints) and consolidating overlapping rules to select the most broadly applicable instructions. This process ensures that every instruction in our taxonomy is a widely recognized engineering standard.

---

> ### Author Response · Authors · 2025-11-24
> **Response to Reviewer sume (Part 2)**
>
> > **W3: The results could be made more actionable by breaking down which categories or instruction types most strongly correlate with functional regressions, and why. This would help identify specific model weaknesses and guide targeted improvements.**
>
> We appreciate the reviewer's suggestion to make the results more actionable. To address this, we conduct a breakdown of functional regression sources and a fine-grained analysis of instruction following rates.
>
> **Correlation with Functional Regression.** We analyze the functional regressions observed in Table 2 for the seven top-performing models. The distribution of regression causes is as follows.
>
> - **Logic & Code Patterns (61.97%):** This category is the dominant cause of functional regression. Instructions in this category, such as limiting branch complexity, frequently require structural code refactoring. This significantly increases the risk of introducing functional bugs compared to other constraints.
> - **Coding Style (23.86%) and Documentation (12.47%):** These categories contribute moderately to regression. While they primarily involve surface-level changes, enforcing strict formatting or specific docstring conventions can occasionally disrupt code execution or syntax.
> - **Error Handling (1.13%) and Library Constraints (0.57%):** These categories have a negligible impact on functional correctness, as they typically involve targeted substitutions rather than structural changes.
>
> ### BigVibeBench
>
> | Models | Documentation | Error | Library | Logic | Style | Overall |
> | :--- | :--- | :--- | :--- | :--- | :--- | :--- |
> | Gemini 2.5 Pro | 52.32 | 99.44 | **100.00** | 84.38 | **90.31** | 79.47 |
> | Gemini 2.5 Flash | 60.25 | 96.38 | 97.58 | 81.06 | 76.89 | 75.91 |
> | Claude 4 Opus | 79.35 | 98.89 | **100.00** | 88.73 | 83.26 | **85.60** |
> | Claude 4 Sonnet | 74.16 | 98.61 | 99.19 | 87.07 | 75.46 | 81.28 |
> | GPT 5 | 68.61 | **99.72** | **100.00** | **93.61** | 73.62 | 81.77 |
> | o4 mini | **80.55** | 99.44 | 98.39 | 90.29 | 75.40 | 84.05 |
> | Kimi K2 | 78.44 | 98.33 | **100.00** | 70.91 | 83.94 | 79.14 |
>
> ### LiveVibeBench
>
> | Models | Documentation | Error | Logic | Style | Overall |
> | :--- | :--- | :--- | :--- | :--- | :--- |
> | Gemini 2.5 Pro | 53.90 | **100.00** | 76.49 | **96.52** | 76.87 |
> | Gemini 2.5 Flash | 54.18 | **100.00** | 68.06 | 79.24 | 68.13 |
> | Claude 4 Opus | 67.46 | **100.00** | 81.25 | 84.34 | 78.75 |
> | Claude 4 Sonnet | 69.34 | 98.00 | 74.79 | 80.11 | 75.20 |
> | GPT 5 | 76.41 | **100.00** | **84.69** | 85.21 | **82.82** |
> | o4 mini | **78.28** | 92.00 | 68.28 | 77.07 | 73.82 |
> | Kimi K2 | 73.23 | 96.00 | 49.62 | 77.44 | 64.76 |
>
> **Fine-grained Model Weaknesses and Strengths.** We further conduct a fine-grained evaluation of IF rates across different instruction categories for the top-performing models in the single-turn setting with five instructions. Since LiveVibeBench contains negligible Library instructions, we exclude that category for that benchmark. Our analysis reveals distinct performance signatures for different models.
>
> - **Universal Competence:** All top models demonstrate near-perfect adherence to Error Handling and Library Usage instructions, with success rates approaching 100%.
> - **Model-Specific Strengths:** Different models exhibit clear specializations. Gemini 2.5 Pro excels particularly at Coding Style. GPT-5 demonstrates superior performance in Logic & Code Patterns. o4 mini performs best on Documentation tasks. Claude 4 Opus exhibits no significant weaknesses across any category.
> - **Consistency:** These performance signatures remain consistent across both real-world tasks (BigVibeBench) and algorithmic problems (LiveVibeBench).
>
> These insights provide actionable guidance for model selection based on specific user needs (e.g., choosing Gemini for style-strict projects or GPT-5 for logic-heavy tasks) and highlight targeted areas for future model development.

---

> ### Author Response · Authors · 2025-11-24
> **Response to Reviewer sume (Part 3)**
>
> > **Q1: During benchmark construction, the paper states that an LLM was used to select relevant and non-conflicting instructions. How was this LLM-based selector evaluated for accuracy and reliability?**
>
> **Human Inspection Results.** We appreciate the reviewer for pointing out this missing detail. To address this, we conduct a manual spot-check on 100 randomly sampled instances (50 from BigVibeBench and 50 from LiveVibeBench), totaling 500 selected instructions. Two authors independently review these instructions for relevance and non-conflict. The review yields a **100% validity rate** on both dimensions.
>
> **Explanation of High Reliability.** We explain the high reliability across these two dimensions as follows.
>
> - **Relevance:** The majority of instructions in our taxonomy, such as Coding Style, Logic & Code Patterns, and Documentation, are universally applicable to almost any code generation task. For context-dependent instructions (e.g., library constraints for file operations), current LLMs (Gemini 2.5 Pro and Claude 4 Opus) are capable enough to identify whether the user query involves relevant operations and filter accordingly.
> - **Non-Conflict:** Direct conflicts between instructions are rare and explicit (e.g., enforcing "Google-style docstrings" vs. "No docstrings," or conflicting parameters). LLMs demonstrate strong capabilities in detecting and resolving these logical contradictions during selection.
>
> **Contrast with Parameter Selection.** In fact, the primary challenge we observe is not the selection logic itself, but Parameter Selection (generating valid values for templates), which exhibits a low invalid rate of approximately 1% to 3%. As detailed in Section 3.1, we explicitly address this issue via rule-based validation in our pipeline.
>
> ---
>
> We hope that our clarifications and additional experiments have resolved your concerns. We thank you again for the constructive feedback, which has helped us improve the clarity and rigor of our work. We are happy to answer any further questions!

---

### Official Review · Reviewer_GhN4 · 2025-10-30

**Soundness:** 3
**Presentation:** 4
**Contribution:** 3
**Rating:** 4
**Confidence:** 4

**Summary:**

Vibe Checker brings an important question to the table---is evaluating pass@1 (functional correctness) sufficient for coding models? The obvious answer is that it is not. Vibe Checker provides some initial evidence and methodology indicating how to convert benchmarks into a new benchmark that checks for coding style and other points of interest.

**Strengths:**

The problem is not original, but the quality and clarity of the paper is high. Additionally, this is an important aspect of code generation and I believe the paper would have high impact, especially if it can be extended to other benchmarks.

**Weaknesses:**

The biggest weakness is that the results seem to naturally conflict with each other / hold no trend.
For example, in single instruction some models (claude) seem to *improve* on functional testing with extra instructions.
Another example is that the regression trend sseems to be much more significant for livevibebench which, according to the paper, is the *less* realistic benchmark problems.
There's also no explanation provided in the paper as to *why* these phenomena are occurring.
The paper could explain why some models are better than others / what the errors look like.

An additional weakness is that there's no statistical significance in any of the results. it's unclear if any of the regressions are meaningful and the coloring for percentages seem fairly arbitrary.

**Questions:**

In addition to answering the points under weaknesses, how easily can this extend to other benchmarks?

---

> ### Author Response · Authors · 2025-11-24
> **Response to Reviewer GhN4 (Part 1)**
>
> We thank the reviewer for recognizing the high quality and clarity of our paper, and for highlighting the potential high impact of Vibe Checker. We appreciate your constructive feedback on interpreting the results. Below, we address your concerns regarding the result trends and experimental details.
>
> > **W1: The biggest weakness is that the results seem to naturally conflict with each other / hold no trend. For example, in single instruction some models (claude) seem to improve on functional testing with extra instructions.**
>
> Thank you for highlighting the specific behavior of the Claude model! We clarify that the overall trends are consistent across the broader model landscape, with Claude 4 Opus being a notable outlier.
>
> **General Trends across Model Families.** Using BigVibeBench as an example, Claude 4 Opus is the only model among the 31 evaluated LLMs that consistently avoids functional regression as the number of instructions increases. The other 30 LLMs all exhibit regression, and the trend is almost strictly monotonic: more instructions lead to larger functional drops. We apologize that Table 2 only lists a subset due to space constraints; however, **Appendix Table 5** provides the complete results where this pattern is evident. We further provide analysis and additional visualizations as follows:
>
> - **Aggregated Evidence:** Figure 3 in our paper illustrates this clear correlation across the averaged performance of all evaluated models: as the number of instructions increases, functional regression rises, and IF rates drop. Regarding interaction modes, single-turn generation consistently preserves functionality better, whereas multi-turn editing consistently achieves higher IF scores.
> - **Family-level Evidence:** To further demonstrate this consistency, we provide an **[anonymous link](https://anonymous.4open.science/r/Trending/README.md)** visualizing performance trends for four major model families: Gemini, Claude, OpenAI, and Qwen. These plots confirm that across these families, increasing constraints consistently leads to higher functional regression and lower IF scores.
>
> **Outlier Analysis (Claude 4 Opus).** Claude 4 Opus is indeed unique. As detailed in Appendix Table 5, it stands out by maintaining functional integrity under multiple instructions on BigVibeBench. We hypothesize that Claude 4 Opus possesses specific optimizations for instruction following that allow it to better manage the trade-off between functionality and constraints. This observation aligns with our response in **W3**, which indicates that Claude 4 Opus exhibits no significant weaknesses across different instruction categories compared to other models.
>
> ---
>
> > **W2: The regression trend seems to be much more significant for livevibebench which, according to the paper, is the less realistic benchmark problems. There's also no explanation provided in the paper as to why these phenomena are occurring.**
>
> We attribute the higher functional regression on LiveVibeBench to two primary factors: the distribution of instruction categories and the nature of the underlying training data.
>
> - **Impact of Instruction Categories.** Our detailed error analysis reveals that 61.97% of functional regressions are caused by the "Logic & Code Patterns" category. This is because instructions in this category (e.g., limiting branch complexity) frequently require structural code refactoring, which increases the risk of introducing functional bugs. As shown in Figure 12 (Appendix C.1), LiveVibeBench features a higher proportion of Coding Logic instructions (42.3%) compared to BigVibeBench (35.9%). This higher density of logic-heavy instructions naturally drives the more severe regression observed in the algorithmic benchmark.
>
> - **Data Distribution Gap.** We also attribute this to the discrepancy between the benchmark tasks and the models' training data. BigVibeBench focuses on real-world programming tasks, where non-functional instructions like documentation and style are natural components of the workflow. In contrast, LiveVibeBench consists of algorithmic contest problems. In the context of competitive programming, solutions typically prioritize raw speed and correctness over industrial style or documentation. Consequently, LLMs likely encounter far fewer training data that combine "complex algorithmic logic" with "strict industrial constraints." This makes the task combination in LiveVibeBench more out-of-distribution for the models, leading to higher instability and regression.

---

> ### Author Response · Authors · 2025-11-24
> **Response to Reviewer GhN4 (Part 2)**
>
> > **W3: The paper could explain why some models are better than others / what the errors look like.**
>
> We appreciate the reviewer's suggestion to deepen the performance analysis. To address this, we conduct a fine-grained evaluation of IF rates across different instruction categories for the top-performing models in the single-turn setting with five instructions. Since LiveVibeBench contains negligible Library instructions, we exclude that category for that benchmark.
>
> ### BigVibeBench
>
> | Models | Documentation | Error | Library | Logic | Style | Overall |
> | :--- | :--- | :--- | :--- | :--- | :--- | :--- |
> | Gemini 2.5 Pro | 52.32 | 99.44 | **100.00** | 84.38 | **90.31** | 79.47 |
> | Gemini 2.5 Flash | 60.25 | 96.38 | 97.58 | 81.06 | 76.89 | 75.91 |
> | Claude 4 Opus | 79.35 | 98.89 | **100.00** | 88.73 | 83.26 | **85.60** |
> | Claude 4 Sonnet | 74.16 | 98.61 | 99.19 | 87.07 | 75.46 | 81.28 |
> | GPT 5 | 68.61 | **99.72** | **100.00** | **93.61** | 73.62 | 81.77 |
> | o4 mini | **80.55** | 99.44 | 98.39 | 90.29 | 75.40 | 84.05 |
> | Kimi K2 | 78.44 | 98.33 | **100.00** | 70.91 | 83.94 | 79.14 |
>
> ### LiveVibeBench
>
> | Models | Documentation | Error | Logic | Style | Overall |
> | :--- | :--- | :--- | :--- | :--- | :--- |
> | Gemini 2.5 Pro | 53.90 | **100.00** | 76.49 | **96.52** | 76.87 |
> | Gemini 2.5 Flash | 54.18 | **100.00** | 68.06 | 79.24 | 68.13 |
> | Claude 4 Opus | 67.46 | **100.00** | 81.25 | 84.34 | 78.75 |
> | Claude 4 Sonnet | 69.34 | 98.00 | 74.79 | 80.11 | 75.20 |
> | GPT 5 | 76.41 | **100.00** | **84.69** | 85.21 | **82.82** |
> | o4 mini | **78.28** | 92.00 | 68.28 | 77.07 | 73.82 |
> | Kimi K2 | 73.23 | 96.00 | 49.62 | 77.44 | 64.76 |
>
> **Observations: Performance Profiles and Model Specialization.** Our analysis reveals distinct performance signatures for different models.
>
> - **Universal Competence:** All top models demonstrate near-perfect adherence to Error Handling and Library Usage instructions, with success rates approaching 100%.
> - **Model-Specific Strengths:** Different models exhibit clear specializations. Gemini 2.5 Pro excels particularly at Coding Style. GPT-5 demonstrates superior performance in Logic & Code Patterns. o4 mini performs best on Documentation tasks. Claude 4 Opus exhibits no significant weaknesses across any category. It ranks first on BigVibeBench and second on LiveVibeBench overall. This balanced profile likely contributes to its superior functional stability discussed in W1.
> - **Consistency:** These performance signatures remain consistent across both real-world tasks (BigVibeBench) and algorithmic problems (LiveVibeBench).
>
> These insights provide actionable guidance for model selection based on specific user needs (e.g., choosing Gemini for style-strict projects or GPT-5 for logic-heavy tasks) and highlight targeted areas for future model development.
>
> Regarding error patterns, we observe diverse failure modes rather than a single consistent pattern across models. To facilitate further community analysis, we will make every effort to release the full set of generated outputs from all 31 LLMs. This resource allows researchers to conduct more granular qualitative studies on specific failure cases and model behaviors.

---

> ### Author Response · Authors · 2025-11-24
> **Response to Reviewer GhN4 (Part 3)**
>
> > **W4: An additional weakness is that there's no statistical significance in any of the results. it's unclear if any of the regressions are meaningful and the coloring for percentages seem fairly arbitrary.**
>
> Thank you for raising this point regarding statistical significance. Our experimental design incorporates several measures to ensure robustness: we utilize a large-scale dataset comprising over 2,000 instances and 10,000 instructions for evaluation. Furthermore, we conduct experiments using deterministic or low temperatures (0.0 and 0.2) across a diverse set of 31 models to ensure our conclusions are comprehensive and applicable.
>
> ### Single-Turn Generation
>
> | Models | 0 Inst | 1 Inst | 2 Inst | 3 Inst | 4 Inst | 5 Inst |
> | :--- | :--- | :--- | :--- | :--- | :--- | :--- |
> | Gemini 2.5 Pro | 50.48 ± 0.22 | 50.11 ± 0.25 | 49.06 ± 0.16 | 49.95 ± 0.18 | 50.41 ± 0.29 | 49.62 ± 0.16 |
> | Gemini 2.5 Flash | 47.38 ± 0.19 | 46.90 ± 0.25 | 46.74 ± 0.31 | 46.24 ± 0.18 | 46.62 ± 0.28 | 46.22 ± 0.08 |
> | Gemini 2.0 Flash | 48.37 ± 0.18 | 47.30 ± 0.22 | 48.14 ± 0.37 | 47.72 ± 0.19 | 46.79 ± 0.26 | 46.08 ± 0.35 |
> | Gemini 2.0 Flash Lite | 46.95 ± 0.24 | 44.65 ± 0.41 | 43.49 ± 0.07 | 43.58 ± 0.25 | 43.62 ± 0.11 | 42.89 ± 0.19 |
>
> ### Multi-Turn Editing
>
> | Models | 0 Inst | 1 Inst | 2 Inst | 3 Inst | 4 Inst | 5 Inst |
> | :--- | :--- | :--- | :--- | :--- | :--- | :--- |
> | Gemini 2.5 Pro | 50.48 ± 0.22 | 49.50 ± 0.08 | 49.09 ± 0.12 | 48.35 ± 0.15 | 47.95 ± 0.18 | 47.79 ± 0.06 |
> | Gemini 2.5 Flash | 47.38 ± 0.19 | 47.28 ± 0.29 | 46.94 ± 0.11 | 46.81 ± 0.09 | 46.69 ± 0.07 | 45.92 ± 0.14 |
> | Gemini 2.0 Flash | 48.37 ± 0.18 | 47.00 ± 0.13 | 45.99 ± 0.10 | 45.24 ± 0.06 | 44.41 ± 0.19 | 43.89 ± 0.16 |
> | Gemini 2.0 Flash Lite | 46.85 ± 0.24 | 45.41 ± 0.35 | 44.75 ± 0.15 | 44.40 ± 0.11 | 42.92 ± 0.08 | 42.77 ± 0.12 |
>
> **Statistical Significance Analysis.** To further address the concern regarding result stability, we conduct repeated experiments (five runs) on the Gemini model family using BigVibeBench. The results, showing mean pass rates and standard deviations, yield the following takeaways.
>
> - **High Stability:** Overall, the results remain highly stable under our inference settings and data size. The standard deviations are consistently low (mostly between 0.05 and 0.35), confirming that the functional regressions we report are statistically significant and not artifacts of random fluctuations.
> - **Interaction Mode Stability:** Multi-turn editing exhibits even lower variance compared to single-turn generation. We attribute this to the richer context provided by previous code generations, which effectively stabilizes the model's output distribution.
>
> These findings confirm that our conclusions are robust and reproducible.
>
> **Presentation and Coloring.** Regarding the coloring for percentages, we clarify that the highlighting in the main text tables serves to improve scannability and help readers quickly identify key trends. We also provide full uncolored tables for all models in the Appendix (Tables 5 to 10), which allow exact inspection without any visual thresholding.
>
> ---
>
> > **Q1: In addition to answering the points under weaknesses, how easily can this extend to other benchmarks?**
>
> Extending our framework to other benchmarks is seamless and straightforward. We distinguish between two scenarios.
>
> **Extending to Python benchmarks.** For other Python-centric benchmarks, the process is effectively "plug-and-play." Since our VeriCode taxonomy already comprehensively covers modern Python industrial standards, we simply apply our LLM selector to the new benchmark's prompts to inject relevant verifiable instructions.
>
> **Extending to other languages.** For benchmarks in other languages, our framework provides a streamlined and universal pipeline for adaptation. While this involves defining a language-specific taxonomy, the process is fully standardized. The same pipeline applies by sourcing candidate rules from that language’s standard linters and industrial style guides, filtering them automatically, and implementing deterministic checkers. For example, this could be done with ESLint for JavaScript or Checkstyle for Java. After that instantiation, the benchmark augmentation procedure remains exactly the same.
>
> ---
>
> We thank the reviewer for the thorough review and for providing such constructive feedback. We have carefully conducted additional experiments to address your concerns, as detailed in our response. Please let us know if you have any further questions!

---

### Official Review · Reviewer_hQF5 · 2025-10-31

**Soundness:** 3
**Presentation:** 3
**Contribution:** 2
**Rating:** 4
**Confidence:** 3

**Summary:**

The paper proposes a new framework for evaluating large language models in code generation. The authors argue that current benchmarks only focus on functional correctness. At the same time, they propose to evaluate both functionality and instruction following. First, they introduce the VERICODE, a set of 30 verifiable code instructions across five categories (coding style, logic, documentation, error handling, and API usage) with corresponding deterministic verifiers. Then, they construct VIBECHECKER from current benchmarks(BigCodeBench and LiveCodeBench) to access functional correctness and instruction following in single-turn and multi-turn code generation. Experiment on 31 LLMs illustrating that non-functional instructions cause functional regression; multi-turn editing improves instruction following but reduces functional accuracy; and a mixture of functional and instruction following metrics best correlates with human preferences.

**Strengths:**

1. The paper is timely and articulates "vibe check" as a measurable composite of functionality and instruction following.

2. The paper augments existing benchmarks, including BigCodeBench and LiveCodeBench, with verifiable instructions to better simulate real-world interactions.

3. The method is carefully designed and curated in three stages, including sourcing a candidate pool, multi-stage filtering, and review and verification to ensure the reach of developers' expectations.

4. The experiment spanned 31 LLMs from 10 model families in two settings, including single-run generation and multi-turn editing. Providing a comprehensive evaluation across interaction contexts.

**Weaknesses:**

1. Honestly, I am not fully convinced by the motivation behind this work. I do not believe coding should be a subjective matter. Code is a tool, and the most important aspect is whether it achieves its intended purpose and all of these can be measured by test cases. Talking back to the example in Figure 1, the choice between using a for loop or recursion is not the goal itself. The critical point is that recursion may exceed memory limits in certain cases (so programmers may not want to use it). All of this can be verified through test cases, because it is simple to add assertion conditions in the test code (e.g., memory ≤ limit). I do not believe there is a specific group of people who inherently prefer a particular coding style. If there is, they are not programmer but artists. This is increasingly obvious in "vibe coding" stage, because there is an increasing amount of AI-generated code and less needs for humans to read all the code. The only aspect of "vibe" that I believe truly matters is readability. However, it is already well discussed in software engineering and has its own metrics (e.g., the Flesch-Kincaid score).

2. Line 180 mentioned that the framework is language-agnostic, but the experiment only focuses on Python.

3. Line 207 mentioned the instruction selection uses an LLM-based selector, but no human inspection was discussed.

4. The taxonomy is not generalizable. There can be numerous target vibes in practice.

5. The writing style is not like ML paper (this is more suitable to submit to HCI conferneces).

**Questions:**

How was the accuracy for instruction selection?

---

> ### Author Response · Authors · 2025-11-24
> **Response to Reviewer hQF5 (Part 1)**
>
> We thank the reviewer for acknowledging our work’s timeliness, comprehensive evaluation, and careful benchmark design. We appreciate the constructive feedback. Below, we address the concerns regarding motivation and specific technical details.
>
> > **W1: I am not fully convinced by the motivation behind this work. I do not believe coding should be a subjective matter. Code is a tool, and the most important aspect is whether it achieves its intended purpose and all of these can be measured by test cases.**
>
> We fully agree with the reviewer that code is an objective tool and that its quality should be measured by whether it achieves its intended purpose via rigorous test cases. Our work does not argue that coding quality is subjective. Instead, we argue that objective code evaluation should also include non-functional test cases from industrial standards, because functional metrics alone are currently misaligned with real-world developer preferences.
>
> **Motivation (Empirical Misalignment).** Our motivation comes from an observation that functional-only benchmark rankings can disagree substantially with what developers actually prefer and adopt in practice. A salient example is the Claude family, which is widely favored in real development workflows but does not rank at the top on popular functional benchmarks. Concretely, Claude 4 Opus scores 68.72 on LiveCodeBench (Table 2), yet it is ranked #1 on Coding LMArena with a rating of 1481 based on about 800K human votes (Table 4, w/ SC). In contrast, o4 mini scores 80.95 on LiveCodeBench but has a materially lower LMArena rating of 1428, ranked beyond 40th.
>
> Ideally, an evaluation framework for code generation should identify the models developers prefer to use in daily workflows. The above mismatch indicates a clear gap between functional metrics and real-world preference. Our work is motivated by understanding this gap and by providing an objective and scalable way to measure an additional dimension that developers care about.
>
> **Our solution: More Test Cases from Objective Industrial Standards for Instruction Following.** To close this gap, we augment standard functional benchmarks with explicit and deterministic instruction-following verifiers derived from industrial coding standards. This adds objective test cases beyond functional correctness. Empirically, we find that our composite score aligns significantly better with human preference than functionality alone, and helps explain why certain models are preferred despite only moderate functional scores.
>
> **Discussion on "Vibe" Aspects.** We respectfully disagree that code style is unimportant or that readability is the only meaningful non-functional aspect. As model capabilities grow and AI-assisted programming becomes prevalent, we expect generated code to be production-ready. This entails adherence to standard styles, maintainability, specific constraint satisfaction, and modern library usage. These aspects are fully covered by our taxonomy, which is grounded in real-world industrial standards. We apologize if Figure 1 caused confusion; the recursion example is merely an illustrative case and is not part of our verifiable taxonomy.
>
> **Positioning in the LLM Evaluation Landscape.** We agree that many non-functional dimensions have long been studied in software engineering. However, evaluation in the LLM code-generation/editing community remains dominated by functional pass@k scores from existing benchmarks, with little systematic measurement of whether models follow explicit developer constraints. By importing established software engineering standards into scalable deterministic verifiers, we aim to complement functional testing and move code evaluation toward better alignment with real-world developer utility.
>
> ---
>
> > **W2: Line 180 mentioned that the framework is language-agnostic, but the experiment only focuses on Python.**
>
> **Python-Centric Experimental Choice.** Our choice to focus on Python experiments is driven by the current code evaluation landscape, which is highly Python-centric. The major benchmarks that are most widely reported and compared in recent LLM literature, including BigCodeBench, LiveCodeBench, and SWE-Bench, are Python-only. Using these benchmarks allows us to demonstrate the practical value of our framework in the setting that the community currently focuses on, while keeping the task distribution directly comparable to prior work.
>
> **Language-Agnostic Framework Design.** On the other hand, we emphasize that our framework design is inherently language-agnostic. Every major programming language possesses established industrial standards and linter ecosystems. Our pipeline of sourcing rules from these standards, applying automated and manual filtering, and implementing deterministic verifiers can be universally applied to any language (e.g., using ESLint for JavaScript or Checkstyle for Java) to construct taxonomies and instructions.

---

> ### Author Response · Authors · 2025-11-24
> **Response to Reviewer hQF5 (Part 2)**
>
> > **W3: Line 207 mentioned the instruction selection uses an LLM-based selector, but no human inspection was discussed.**
>
> **Human Inspection Results.** We appreciate the reviewer for pointing out this missing detail. To address this, we conduct a manual spot-check on 100 randomly sampled instances (50 from BigVibeBench and 50 from LiveVibeBench), totaling 500 selected instructions. Two authors independently review these instructions for relevance and non-conflict. The review yields a **100% validity rate** on both dimensions.
>
> **Explanation of High Reliability.** We explain the high reliability across these two dimensions as follows.
>
> - **Relevance:** The majority of instructions in our taxonomy, such as Coding Style, Logic & Code Patterns, and Documentation, are universally applicable to almost any code generation task. For context-dependent instructions (e.g., library constraints for file operations), current LLMs (Gemini 2.5 Pro and Claude 4 Opus) are capable enough to identify whether the user query involves relevant operations and filter accordingly.
> - **Non-Conflict:** Direct conflicts between instructions are rare and explicit (e.g., enforcing "Google-style docstrings" vs. "No docstrings," or conflicting parameters). LLMs demonstrate strong capabilities in detecting and resolving these logical contradictions during selection.
>
> **Contrast with Parameter Selection.** In fact, the primary challenge we observe is not the selection logic itself, but Parameter Selection (generating valid values for templates), which exhibits a low invalid rate of approximately 1% to 3%. As detailed in Section 3.1, we explicitly address this issue via rule-based validation in our pipeline.
>
> ---
>
> > **W4: The taxonomy is not generalizable. There can be numerous target vibes in practice.**
>
> **Clarifying the Goal of the Taxonomy.** We respectfully clarify that our taxonomy is not intended to decompose an arbitrary or subjective notion of "vibe." Our work follows a logical progression.
>
> - **Observation:** We identify a significant gap where functional benchmark scores fail to align with real-world developer preferences.
> - **Hypothesis:** We hypothesize that the missing piece explaining this gap is relevant to the non-functional aspects of the code.
> - **Taxonomy Design:** To measure this, we construct a taxonomy of verifiable code instructions rooted in modern software engineering specifications. Its sole purpose is to evaluate non-functional instruction following in the coding domain.
> - **Validation:** Our subsequent experiments confirm that instruction following is a critical component of human preference.
>
> Therefore, our taxonomy is unrelated to subjective "numerous target vibes," but stems purely from objective industrial standards. This aligns with the reviewer's feedback that code is objective. We do not aim to cover all arbitrary subjective tastes, but strictly adhere to widely adopted standards to evaluate instruction following.
>
> **Discussion on Generalizability.** Regarding generalizability, we address both intra-language and cross-language aspects.
>
> Within Python, our taxonomy provides sufficient coverage by sourcing candidates from 800+ rules from industrial standards (e.g., Flake8, isort, pycodestyle), and by utilizing parameterized instructions to expand into hundreds of constraints. For cross-language contexts, while specific rules vary, our framework design is fully transferable. The pipeline of extracting rules from standard linters and implementing deterministic verifiers constitutes a universal method for constructing taxonomies in any programming language.
>
> ---
>
> > **W5: The writing style is not like ML paper (this is more suitable to submit to HCI conferneces).**
>
> **Justification of the Writing Style.** Our use of the term "Vibe Check" is a deliberate choice to align our evaluation framework with the emerging real-world workflow known as "vibe coding". We aim to bridge the gap between this user-centric intuition and metric-based machine learning evaluation. By using this terminology, we highlight that the "subjective feeling" of coding often corresponds to concrete, measurable engineering constraints that LLM evaluations currently under-measure.
>
> **Relevance to ML Venues.** We would also like to clarify that the substance of the work is tightly aligned with ML conference topics. The paper focuses on constructing a verifiable instruction dataset, benchmarking LLM performance on large-scale testbeds, and analyzing correlations between automated metrics and human preference. These are core ML contributions on benchmarking, evaluation, and human-aligned model assessment, which fit well within ICLR’s scope, especially the datasets and benchmarks track that we are submitting to.

---

> ### Author Response · Authors · 2025-11-24
> **Response to Reviewer hQF5 (Part 3)**
>
> > **Q1: How was the accuracy for instruction selection?**
>
> As detailed in our response to W3, we evaluate the selection quality through a manual spot-check on 100 randomly sampled instances, totaling 500 selected instructions. The results confirm a 100% validity rate for both relevance and non-conflict dimensions.
>
> ---
>
> We thank the reviewer again for constructive feedback and for raising such insightful discussion points. We hope we have carefully addressed all the points; please let us know if you have any further questions!

---

### Author Response · Authors · 2025-12-02
**Summary of Rebuttal**

Dear Area Chair and Reviewers,

We are deeply grateful for the reviewers' insightful feedback and constructive criticism. Given the recent changes in the review process, we provide this summary to assist the Area Chair in navigating our detailed responses and the additional data we have provided in the discussion threads.

---

**Summary of Strengths.** We appreciate that the reviewers recognized the timeliness, rigor, and potential impact of our work:

- **Timeliness & Impact:** The paper addresses a critical gap in code evaluation by articulating "vibe check" as a measurable composite of functionality and instruction following (Reviewer `hQF5`), with high potential impact in the field (Reviewer `GhN4`).
- **Methodological Rigor:** The framework is carefully designed, utilizing a multi-stage curation process grounded in industrial standards (Reviewers `hQF5`, `sume`).
- **Comprehensive Evaluation:** The evaluation scale is extensive, spanning 31 LLMs from 10 model families across single-turn and multi-turn settings. (Reviewers `hQF5`, `sume`)
- **Presentation:** The paper is of high quality, clear, and well-written. (Reviewer `GhN4`)

---

**Summary of Clarifications and Additional Experiments.** We have provided detailed responses to each reviewer’s specific questions in the threads below. Here, we summarize our resolution of the major themes:

**1. Clarifying the Motivation** (Response to `hQF5`)

We fully agree with the reviewer that code is an objective tool. While our motivation arises from the empirical misalignment between functional benchmarks and human preference, our methodology is strictly grounded in objective industrial standards. Through our testbed, we derive composite metrics for functional correctness and instruction following that are objective and deterministically measurable, yet exhibit the highest correlation with subjective human preference. Essentially, we advance evaluation by decomposing subjective real-world preferences into concrete signals that can be measured and optimized.

**2. Statistical Significance and Quality Assurance** (Response to `GhN4`, `hQF5`, `sume`)

- **Statistical Significance:** We provide repeated experimental results (5 runs) showing very low standard deviations (mostly between 0.05 and 0.35), which confirms that the functional regressions we report are statistically significant and not random noise.

- **Quality Control:** We detail our manual review process, and conduct an additional spot-check of 500 instructions yields a 100% validity rate for relevance and non-conflict, ensuring the reliability of our LLM-based selector.

**3. Analysis of Performance Trends** (Response to `GhN4`)

From our experiments, 30 of the 31 evaluated models exhibit clear functional regression as instruction complexity increases. Furthermore, we provide aggregated evidence and additional family-level visualizations (covering Gemini, Claude, OpenAI, and Qwen) to confirm the robustness of our conclusions.

**4. Deep Analysis and Actionable Insights** (Response to `GhN4`, `sume`)

We provide a fine-grained breakdown of regression causes and model profiling:

- **Root Causes:** We pinpoint that 61.97% of functional regressions stem from "Logic & Code Patterns" instructions (which require structural refactoring), whereas style and documentation constraints cause moderate regressions.

- **Model Profiling:** Our supplementary analysis reveals the specific strengths and weaknesses of different models regarding instruction following (e.g., Gemini's strength in Style vs. GPT-5's strength in Logic), offering actionable insights for model selection and future development. We also analyze the outlier behavior of Claude 4 Opus, which maintains balanced performance across categories.

**5. Scope and Generalizability** (Response to `hQF5`, `sume`)

- **Function-Level Focus:** We justify our focus on function-level evaluation as a necessary foundation that aligns with current human preference data (e.g., LMArena) and provides a prerequisite for complex multi-file tasks.

- **Language Agnosticism:** We clarify that while our experiments focus on Python, our pipeline of extracting rules from standard linters is language-agnostic and easily extensible to other languages and benchmarks.

---

We sincerely thank the reviewers again for their time and contributions to improving our work. The detailed responses provided above aim to effectively address all concerns. Given the current restrictions on reviewer interaction, we would appreciate it if the Area Chair takes these clarifications into full consideration.

Best,

The Authors

---

### Meta-Review · Area_Chair_fDjp · 2025-12-09

**Summary:**

This paper augment BigCodeBench and LiveCodeBench with verifiable code instructions to obtain BigVibeBench and LiveVibeBench. In addition to evaluate pass@k performance, the augmented dataset can also evaluate the none functional part of the generated code.

The major concern is on the motivation and representativeness of the benchmark for the vibe coding setup, where primary focus is to do project level development rather than function level. Additionally, this work did not evaluate LLM based agents (e.g. react or code act based agents), which are the main method used for vibe coding.

**Reviewer Concerns:**

The major concern is on the motivation and representativeness of the benchmark for the vibe coding setup, where primary focus is to do project level development rather than function level. In addition, though authors mentioned that the approach would generalize for other languages, only Python problems were used.

**Reviewer Scores:**

hQF5 4
GhN4 4
sume 4

---

### Decision · Program_Chairs · 2026-01-26

Reject